# The Integration of Azure Sphere and Azure Cloud Services for Internet of Things

**Jiong Shi ***, **Liping Jin and Jun Li**

School of Electronic and Computer Engineering, Zhejiang Wanli University, Ningbo 315100, China
* Correspondence: dearsj001@gmail.com; Tel.: +86-574-88223268

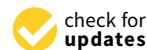

**Featured Application: The proposed integrated solution for Internet of Things (IoT) in this article, which utilizes the fully integrated, high performance Azure Sphere microcontroller unit (MCU), as well as Microsoft Azure cloud services, can be used to meet the requirements of robust end-to-end security user scenarios with limited budgets.**

**Abstract:** Internet of Things (IoT) has become one of the key factors that enables, drives and accelerates the digital transformation all over the world. The vision of the IoT is not only the immediate access to the data but also the ability to turning data into intelligence. As such, there is a growing number of public cloud computing providers offering IoT related services, including data processing, data analyzing and data visualization. However, with tens of billions of microcontroller-powered devices getting involved in the era of IoT, the concerns for overall security, privacy and cost are rising constantly and exponentially. Furthermore, these issues cannot be solved by public cloud computing providers since they mainly focus on the software and services rather than on the end devices. In this article, an integrated solution including Azure Sphere devices and Azure cloud services is proposed to provide a comprehensive and efficient way to ensure security that starts in the device and extends to the cloud with limited budgets. Moreover, the implementation details including hardware components, software design and Azure cloud integration are presented to demonstrate the feasibility and efficiency of the proposed solution.

**Keywords:** Internet of things; Azure Sphere; Azure cloud services; IoT Hub; ZigBee; WSN

## 1. Introduction

With the rapid evolution of the Internet of Things (IoT), more and more network-enabled devices are involved to pave the way for connected vehicles [1], industrial IoT [2,3], connected healthcare [4,5], smart cities [6] and smart home [7] from an infrastructure perspective. Meanwhile, the considerations have been shifted gradually from the real-time data stream to the valuable intelligence derived from the data. Hence, more and more public cloud computing providers, such as Amazon, Microsoft, Google and IBM, are involved in IoT era to provide a variety of services including data aggregation, data filtering, data storage, data processing and data visualization for IoT support [8].

However, there are still some issues and challenges that cannot be neglected in face of ever-growing end devices in large-scale IoT systems. Firstly, the need to incorporate the high-value security into every low-cost (much cheaper than 10 US dollars) network-connected IoT devices is currently underestimated because of the limited development costs and device capabilities [9,10]. Consequently, these network-connected devices are extremely vulnerable to aggressive assault from network attackers [11]. Obviously, this challenge is not considered by public cloud computing providers since their concentration are mainly put on the software and services to make it simple for developing, deploying and maintaining IoT applications. Secondly, according to the literature [12–17], few existing

solutions considered the security aspects all the way from the device to the cloud. To some extent, most of them have achieved the security and privacy goals in one specific layer or multiple layers, rather than overall system security. Thirdly, as hundreds of thousands of end devices are involved in the IoT system, the device provision, configuration as well as management can hardly be integrated in a secure and efficient way. Hence, a more feasible integrated solution is urgently needed to ensure high-value security from the device layer to the cloud layer.

Based on the current IoT trend, we propose an integrated solution that includes Azure Sphere devices and Azure cloud services to address the challenges of overall security, cost and device management. By taking advantage of the Azure Sphere and Azure cloud services, this integration is capable of maintaining the low-cost requirement. Moreover, a proof of concept application (a remote monitoring and feedback control system), namely its design and implement to validate the proposed integrated solution, is detailed. More specifically, the contributions of this study are the following.

- A new IoT integrated solution including hardware, software and services is proposed based on Azure Sphere device and Azure cloud services to meet the requirements of high security and low cost.
- The hardware prototypes for Azure Sphere device, which act either as a direct connected device or a gateway, are designed and implemented based on the MT3620 development kit.
- The programs for MT3620 and CC2530 ZigBee devices, as well as applications for Windows devices are designed and implemented. Meanwhile, a proof of concept for the proposed integration is demonstrated and verified based on the hardware prototypes and Azure cloud services.

The remainder of this article is structured as follows. The state-of-art of IoT integrated solutions are reviewed in Section 2. The overall design of the proposed integration is provided in Section 3. The implementation details that comprise hardware components, software design and Azure cloud integration are presented in Section 4. In Section 5, the experimental results are shown concerning the device layer, cloud layer and application layer. The discussion of the comparison between the proposed security integration and the existing security solutions is given in Section 6. Finally, the conclusion is summarized in Section 7.

## 2. Related Work

### 2.1. Security Research in IoT Systems

Currently, there are two main research topics involving IoT system's security. The first one is the security schemes and policies for resource-constrained IoT systems, while the second one is the design and implementation of security solutions for IoT applications [8]. The research on the first one is currently directed at four critical areas including access control schemes, anomaly detection, security models and key management to achieve the high levels of security. Firstly, for access control schemes in IoT, a large and intensive research effort is devoted to the access control architecture, the type of keys used to secure the communication channel, the access control channel, and the access control logic [18]. In addition, such access control schemes are widely used in implantable medical devices [19], body area networks [20], smart gird with renewable energy resources [21], smart home [22], and industrial networked systems [23]. Secondly, in order to achieve the automatic identification of resource depletion and unauthorized access, anomaly detection is proposed, which is generally categorized into three different ways, namely, unsupervised anomaly detection [24], supervised anomaly detection [25] and semi-supervised anomaly detection [26]. Thirdly, security model is a critical enabling factor that should be paid attention to for creating a trustworthy and interoperable IoT system. In this sense, security policies including hardware security, data encryption, secure routing, risk assessment, intrusion detection, anti-malware solution, firewall and trust management are usually considered from the perspectives of perception, transportation and application levels when the practical IoT application is designed [27,28]. Lastly, key management is of great importance which involves creating,

renewing, transferring and accounting for cryptographic items in lightweight, resource constrained IoT devices [29]. Policies and services about generation, renewal, discovery, reporting, escrow, rollover, destruction and revocation for keys and certificates are usually highlighted during the design process [30,31].

Meanwhile, the research on the IoT integrated solutions has already taken security into account in different layers. A comparison of several surveyed papers is summarized in Table 1. The security challenges in industrial IoT enabled cyber-physical systems was summarized in [12], which highlighted the security in data mining and big data under Cisco IoT framework. A healthcare monitoring framework was designed and implemented in [13], where signal enhancement, watermarking and other related analytics were used to avoid identity theft. Authentication, privacy encryption and secure packet forwarding were studied in [14], and several strategies were proposed under three-layer IoT architecture as well. An overview of security principles, security challenges and proposed countermeasures was presented in [15]. A summary of security issues about protocols and applications for IoT was given in [16], which was based on five-layer IoT framework. A comprehensive list of vulnerabilities and countermeasures on the three layers including edge nodes, communication and edge computing was provided in [17]. To some extent, these studies have achieved the security and privacy goals in one specific layer or multiple layers.

**Table 1.** A summary of security considerations for IoT solutions.

| Solution | Application Area | IoT Layers | Security Considerations |
|:---:|:---:|:---:|:---:|
| [12] | Industrial IoT | Cisco IoT Framework | Security threats in data mining and big data |
| [13] | Health care | Things and clouds layer | Signal monitoring in health care application |
| [14] | General | Sensing, network and application layer | Authentication, privacy encryption, packet forwarding |
| [15] | General | Perception, network and application layer | Security threats and strategies in three layers |
| [16] | General | Object, object abstraction, service management, application and business layer | Security protocols in IoT |
| [17] | General | Cisco IoT seven-layer model | Security threats and strategies in three layers |

However, to the best of our knowledge, there still remain three challenging issues in real IoT solutions. The first one is that low-cost network-connected devices are extremely vulnerable to network attackers due to the limited development costs and device capabilities. The second one is that very few of the existing integrated solutions have considered the security all the way from the device to the cloud. The third one is that the integration of device provision, configuration as well as management can hardly be achieved in a secure and efficient way. Motivated by this fact, this article is focused on the implementation of an integrated solution that could ensure security for IoT systems, from the low-cost IoT devices to the applications and services deployed in the cloud.

### 2.2. Azure Sphere

In order to deal with this challenging problem, a novel class of price-sensitive application platform called Azure Sphere was released by Microsoft which integrates real-time processing capabilities with a secure, internet-connected operating system [32]. Along with the Azure Sphere MCU SDK tools for application development, the Azure Sphere Security Service is also included for secure cloud and web connection.

The goal of the Azure Sphere is to enable IoT designers to incorporate high levels of security into every low-cost MCU device with network connectivity. Based on the extensive experience in software design for device security, seven necessary properties of highly secure, network-connected devices were identified by Microsoft, namely, hardware-based root of trust, small trusted computing base, defense in depth, compartmentalization, certificate-based authentication, security renewal and failure reporting [33]. Motivated by the referred goals, Azure Sphere MCU, Azure Sphere OS and Azure Sphere Security Service were designed to work together in a harmonious whole to reduce risks. The architecture of the Azure Sphere comprises the following three main components.

- The Azure Sphere MCU is composed of multiple ARM Cortex cores, network connectivity subsystem, multiplexed I/O peripherals, integrated RAM and flash, hardware firewalls, and the Pluton security subsystem, which provide built-in security from a hardware perspective.
- The custom Linux-based kernel, the Security Monitor and OS services that host the application container add a four-layer defense, in-depth secure environment from a software perspective.
- The Azure Sphere Security Service that merges with certificate-based authentication, timely update and failure reporting renews security to confront with emerging threats from a service perspective [34].

With the help of silicon partner MediaTek, the first Azure Sphere certified MCU MT3620 was released at the end of 2017 [35]. The price of MT3620 is less than 8.65 US dollars, which covers the physical MCU chip, licenses for the chip, the Azure Sphere OS and the Azure Sphere Security Service. MT3620 includes both ARM Cortex-A7 application processor and ARM Cortex-M4F I/O subsystems, which are designed for running Azure Sphere OS and real-time control requirements of on-chip peripherals, respectively. In addition to the high-performance ARM cores, the Pluton security subsystem and the on-board Wi-Fi subsystem are embedded in MT3620 to handle secure boot, secure system operation and high throughput network connectivity. For targeting a wide range of IoT applications, five serial interfaces, two I2S interfaces, eight ADC channels, twelve PWM outputs and seventy-two programmable GPIOs are included. Subsequently, in September 2018, the Azure Sphere MT3620 development kit was released by Seeed Studio to support rapid prototyping with Azure Sphere technology [36,37].

## 3. System Design

### 3.1. Overview

The architecture of the proposed integrated solution for IoT is presented in Figure 1. The system is composed of three fundamental modules: (i) the device connectivity and management module; (ii) the data processing and management module; and (iii) the business connectivity and data visualization module.

The module of device connectivity and management is composed of Azure Sphere device, Azure Sphere gateway, existing IoT device, resource constrained device and cloud gateway (Azure IoT Hub).

- Azure Sphere device stands for the MT3620 development kit that directly connects with Azure IoT Hub.
- Existing IoT device and resource constrained device are the legacy IoT devices that cannot establish secure communication with Azure IoT Hub directly. Instead, they communicate with the Azure Sphere gateway to achieve data sending and receiving.
- Azure Sphere gateway acts as the bridge between legacy IoT devices and Azure IoT Hub.
- Azure IoT Hub is the cloud gateway that acts as a central message hub for bidirectional communication between Azure services and the IoT end devices.

In this module, the solution provides two different methods to connect the devices to the cloud gateway according to the communication capability of the device. First of all, relying on the dual-band 802.11a/b/g/n Wi-Fi component, MT3620 is powerful enough to establish a secure direct IP connection to Azure IoT Hub. Secondly, for devices using short-range communication technologies such as ZigBee, Bluetooth, 6LowPAN and other industry standards, MT3620 can be configured as a field gateway for protocol translation and custom processing before the data is uploaded to the Azure IoT Hub. Azure IoT SDK is provided for both scenarios described above to facilitate the access to Azure IoT Hub for MT3620. Azure IoT Hub is utilized as the cloud gateway to provide secure connectivity, event ingestion and device management services for MT3620 devices.

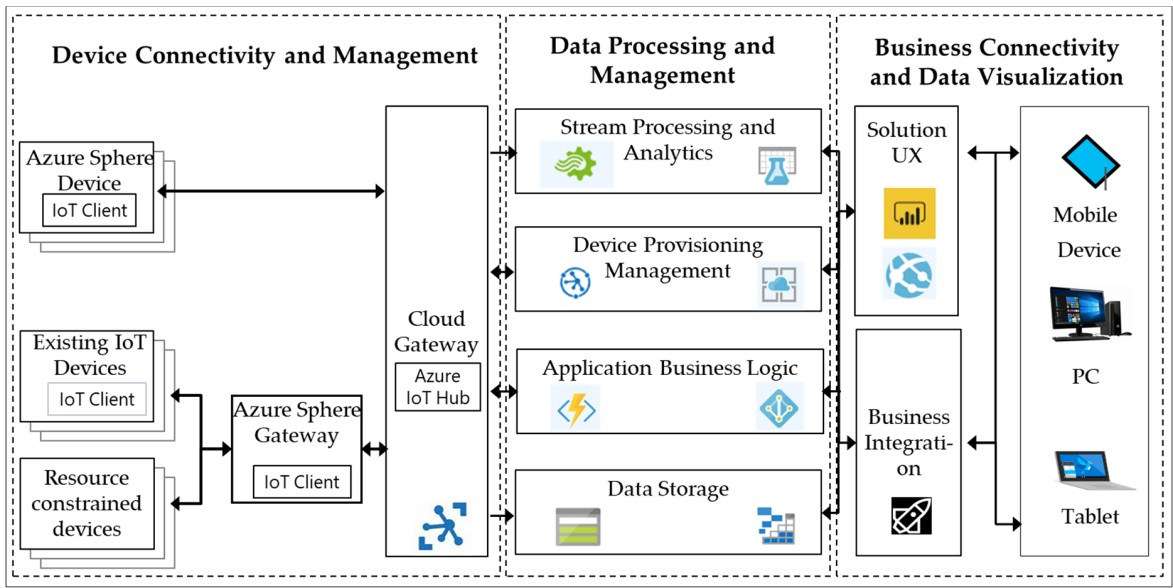

**Figure 1.** System overview.

The module of data processing and management offers stream processing and analytics, device provisioning, application business logic and data storage services. They are explained as follows.

- Stream processing and analytics mainly make use of Azure Stream Analytics service for large-scale data processing with complex rules.
- Device provisioning management functionality is provided by Azure Device Provisioning Service, which allows the registration and connection of large sets of MT3620 devices to Azure IoT Hub for bidirectional communication.
- Application business logic is responsible for action executing based on insights generated from data during stream processing.
- Azure Table Storage service is used for data storage in the cloud.

The module of business connectivity and data visualization includes components which are used for user interface and business integration. In general, the user experience (UX) can be achieved in different devices like mobile phones, traditional PCs and tablets. In this article, Power BI is utilized as a cross-platform tool to provide online and offline data visualization. Native application is designed and developed to run on Windows devices for real-time data visualization and feedback control.

*3.2. Hardware Design*

According to the designed solution, Azure Sphere MT3620 is the essential component of the proposed system. MT3620 is the first Azure Sphere certified MCU, which is embedded with tri-core

microcontroller, one ARM Cortex-A7 core runs at up to 500 MHz as application processor and two ARM Cortex-M4F cores run at up to 200 MHz as general-purpose processor. In the proposed system, the MT3620 development kit released by Seeed Studio is used as Azure Sphere device to support real-time and security requirements when interfacing with UART, I2C, SPI and ADC on-chip peripherals. The MT3620 development kit, powered by rechargeable lithium-ion batteries, is designed to perform as direct connected device or gateway for legacy nodes. For the former scenario, sensors and actuators including temperature and humidity sensor, light sensor, sound sensor, and a relay are connected via on-board interfaces. And direct HTTP or HTTPS message transmission with Azure IoT Hub can be achieved with 2.4/5 GHz dual-band Wi-Fi module. For the latter scenario, resorting to the UART interface, the legacy IoT systems such as ZigBee, Bluetooth and other network devices can be connected to the MT3620 gateway for message processing before it is uploaded to the Azure IoT Hub. In this article, a ZigBee wireless sensor network (WSN) including ZigBee End Device and ZigBee Coordinator is established as legacy IoT system for environmental perception. ZigBee End Device is equipped with temperature and humidity sensor, light sensor, gas sensor, and passive infrared (PIR) sensor.

### 3.3. Azure Cloud Intergration

The services provided by Azure cloud, play a key role in data collecting, data processing, data storage and data visualization. This subsection contains all the services running on Azure cloud.

#### 3.3.1. Azure IoT Hub

The Azure IoT Hub plays a central role. It acts as a bridge between the Azure Sphere devices and the Azure cloud services. On one hand, not only the device-to-cloud messages are collected by Azure IoT Hub to understand the real-time state of the MT3620 devices, but also the cloud-to-device commands and notifications are sent reliably to update the policies of sensor data collecting that are stored in the MT3620 device. On the other hand, a zero-touch, just-in-time provisioning is achieved with the help of Device Provisioning Service, which means that millions of devices can be provisioned in a secure and scalable way with no human intervention.

#### 3.3.2. Azure Stream Analytics

Azure Stream Analytics [38] seamlessly connect Azure IoT Hub with Power BI and Azure Table Storage to enable rich data visualization and persistent data storage. For data visualization, the Azure IoT Hub is selected as data input and Power BI is adopted as the output to help to transform live data into actionable and insightful visual reports. A SQL-like query language, which is a subset of standard T-SQL, is used for real-time streaming computations and filtering. In this solution, all the data collected from Azure IoT Hub is streamed to the output of Power BI, and the query statement is configured as "SELECT * INTO [PowerBI] FROM [AzureSphereData]", in which "AzureSphereData" stands for the input alias, while "PowerBI" represents the data sink. For data storage, the Azure IoT Hub is configured as data input, while Azure Table Storage is selected as data sink.

#### 3.3.3. Data Visualization and Storage

Power BI is a powerful business service for data visualization through interactive, real-time dashboards which is available on a wide range of devices including Windows, iOS and Android. Visual element Gauge is selected as the widget for average temperature and average humidity, while Colum Chart is utilized for visualization of the maximum light and minimum sound. In order to understand the status of the remote device, the real-time sensor values including sound, light, temperature and humidity are displayed on two different Area Charts. Once the editing reports are finished, the appearance will be consistent on all platforms to make a coherent user experience. Azure Table storage is suitable for storing structured NoSQL data in the cloud, such as device information, the user data for web application, address books and so on. The "Azure Sphere Table" is firstly created manually with the Azure Storage Account. The entity of the table contains nine

properties such as temperature, humidity, light and sound. The partition key, row key and time stamp are automatically generated for every entity in the table, which are representative of the first part of an entity's primary key, the second part of an entity's primary key and the time that the entity was last modified.

### 3.4. Configure Azure IoT Hub for Azure Sphere

To use Azure Sphere MT3620 devices with Azure IoT Hub, there are a few tasks that need to be completed. As a prerequisite, an Azure IoT Hub shall be created with a suitable Resource Group and Region on Azure Portal [39]. For a lower latency, the closest location shall be selected. After that, a new instance for the IoT Hub Device Provisioning Service [40] is created and linked with the existing IoT Hub.

The Azure Sphere device provision process is shown in Figure 2. Firstly, Azure Sphere Developer Command Prompt is used to download the Certificate Authority (CA) for the Azure Sphere tenant. Secondly, the tenant CA certificate shall be uploaded to Azure Device Provisioning Service and then a verification code will be generated to validate the certificate ownership. Thirdly, the validation certificate which proves the ownership shall be downloaded by Azure Sphere Developer Command Prompt with the verification code from the step above. Lastly, the certification process is completed as soon as the signed validation certificate is uploaded to the Azure Device Provisioning Service on Azure Portal. Afterwards, the custom Enrollment Group is created with the certificate that was validated in the previous step to add the Azure Sphere device.

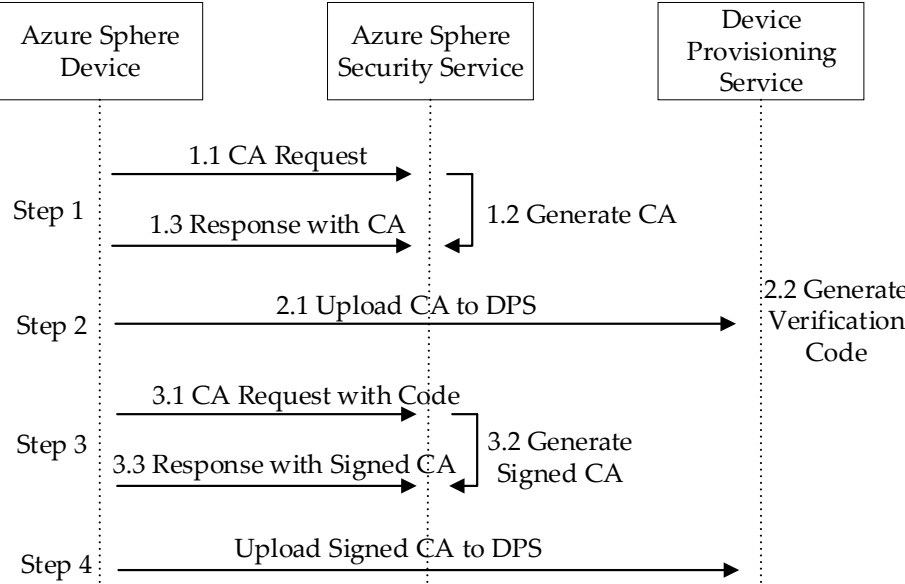

**Figure 2.** Azure Sphere device provision with Security Service and Device Provisioning Service.

By the above configuration, not only the use of compromised and untrusted devices is prevented, but also the registration of the authorized devices in the IoT Hub can be ensured.

### 3.5. System Security

In order to ensure security through all layers of the IoT system, the security policies and rules were divided into four categories as shown in Table 2, i.e., device security, connection security, cloud security [41] and application security.

For device security, the MT3620 MCU used in this system is identified as a critical component that is designed according to the seven highest levels of security properties [33]. Meanwhile, certificate-based authentication, instead of passwords, is utilized to prove identities when communicating with the

cloud servers. More details about authentication process can be found in Section 3.4. Furthermore, to ensure the security of the application and its data, four strategies including limited access to external resources, application capabilities, signing images and device capabilities are implemented.

For connection security, the communications between MT3620 and the cloud gateway are secured by the wolfSSL [42], which is an industry-standard Transport Layer Security (TLS) library targeted at IoT, embedded and RTOS environments. Specifically, HTTP Secure (HTTPS) and Message Queuing Telemetry Transport (MQTT) protocols are used not only for efficient resource usage but also for reliable message delivery.

**Table 2.** Security considerations in this project.

| Categories | Policies and Rules |
|---|---|
| Device security | Seven highest levels of security properties<br>Certificate-based authentication<br>Application and its data Security |
| Connection security | Transport Layer Security<br>HTTPS and MQTT protocols |
| Cloud security | User authentication and authorization<br>ISO 27001 and ISO 27018 certified<br>256-bit AES encryption |
| Application security | 128-bit AES algorithm for ZigBee<br>Application capabilities, device capabilities and signing deployment for MT3620 |

For cloud security, Azure Active Directory (AAD) is used for user authentication and authorization. Azure Stream Analytics is ISO 27001 and ISO 27018 certified [43], which means that information security management and personal data protection in the cloud is ensured. All data written to the Azure Table Storage is encrypted with 256-bit Advanced Encryption Standard (AES) encryption, which is considered as one of the strongest block ciphers available [44].

For application security, the designed solution achieves 128-bit AES algorithm for data encryption between the CC2530 ZigBee nodes. Specifically, the CC2530 chipset takes advantage of AES-CCM (Counter with CBC-MAC) mode to encrypt and decrypt data. The global variables zgPreConfigKeys and DSECURE in the configuration file "f8wConfig.cfg" are set to TRUE and 1, respectively, to enable AES security. Secondly, application capabilities, device capabilities and signing deployment are introduced to ensure the application security running on MT3620. Specifically, application capabilities are defined in application manifest file to declare the authorized use of resources that a given application requires. Device capabilities of the Azure Sphere device are granted by the Azure Sphere Security Service and are stored in flash memory. All image packages deployed to an Azure Sphere device must be signed with an SDK signing key. Before the application is uploaded to the chip by sideloading or over-the-air (OTA) method, the packages should be signed to ensure its security.

## 4. System Implementation

This section presents the detailed implementation of the proposed solution. In the light of the system overview that is depicted in Section 3.1, the proof of concept for the proposed integration is shown in Figure 3.

The entire system can be divided into three different layers including device layer, cloud layer and application layer. It is noted that there are some connections between Figures 1 and 3. Firstly, a ZigBee WSN including ZigBee End Device and ZigBee Coordinator stands for the legacy IoT system in Figure 1. Secondly, MT3620 development kits are used as substitutions for Azure Sphere device and gateway in Figure 1. Thirdly, Azure IoT Hub, Azure Stream Analytics, Azure Function, Azure Device Provisioning Service, Azure Storage Table and Azure Cosmos DB are utilized as the data processing

and management unit in Figure 1. Lastly, Power BI and Native App are used as UX solution and business integration in Figure 1, respectively.

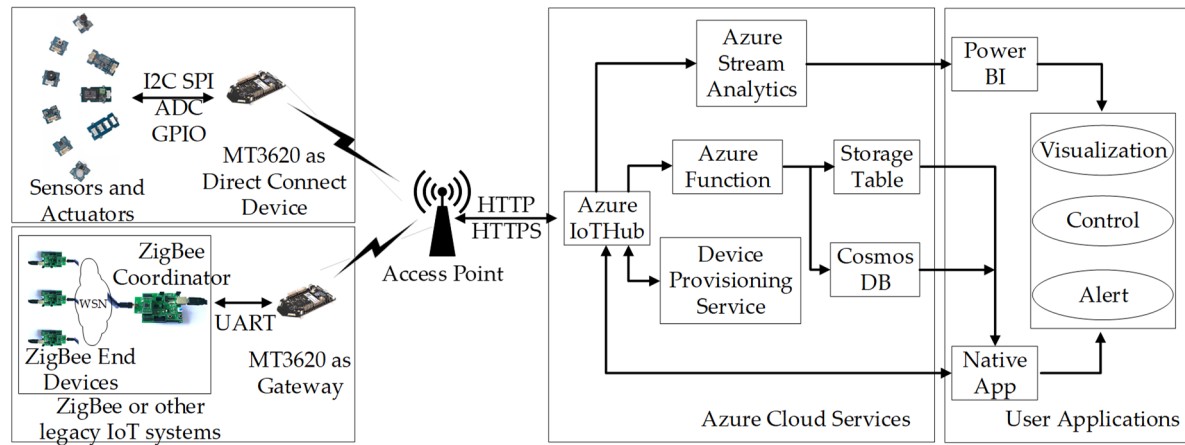

**Figure 3.** Proof of concept for the proposed solution.

*4.1. Hardware*

The designed hardware prototype is given in Figure 4. The MT3620 development kits are designed to perform as direct connected device and gateway device. For the former scenario, sensors and actuators are directly connected to the MT3620 development kit via on-board interfaces. For the latter scenario, ZigBee network device can be connected to the MT3620 field gateway with UART interface. On the system implementation, ZigBee end device is equipped with sensors and actuator as shown in Table 3, including temperature and humidity sensor, light sensor, gas sensor, passive infrared (PIR) sensor and a relay.

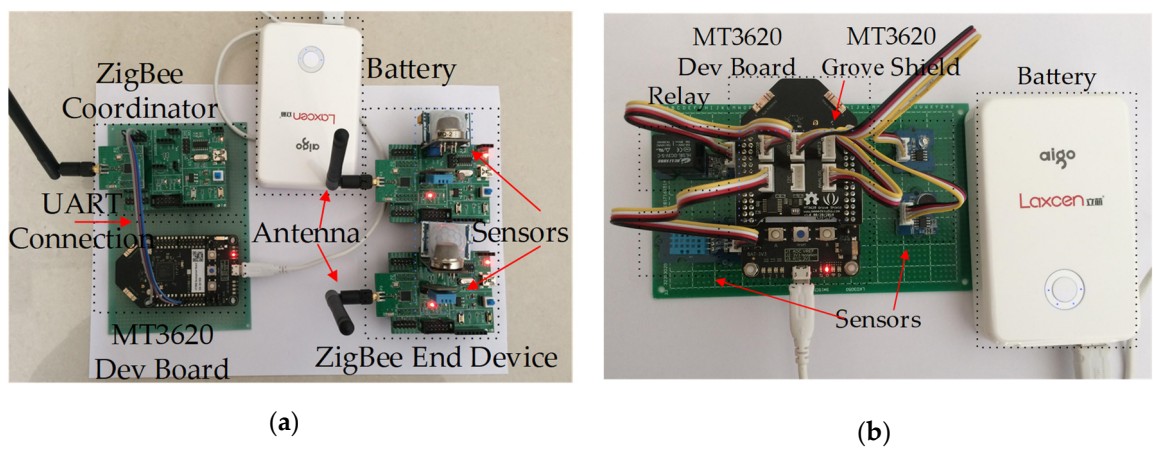

(**a**)      (**b**)

**Figure 4.** Hardware prototypes: (**a**) MT3620 as gateway device; (**b**) MT3620 as direct connect device.

**Table 3.** Sensors and actuator used in this project.

| Sensor | Model | Type |
| --- | --- | --- |
| Temperature and Humidity | Grove [1]-DHT11 | Digital |
| Light | Grove-Light Sensor | Analogue |
| Sound | Grove-Sound Sensor | Analogue |
| Gas | Grove-Gas Sensor (MQ2) | Analogue |
| PIR | Grove-PIR Motion Sensor | Digital |
| Relay | Grove-Relay | Digital |

[1] All sensors are manufactured by Seeed Studio.

For MT3620 as direct connected device, since Azure Sphere SDK has no ADC and does not support I2C communications, the MT3620 Shield is used as an interface between MT3620 and external sensors. By importing MT3620 Grove Shield Library to the project solution, Grove sensors listed in Table 3 are enabled through the functions that are provided in the library.

In addition, there are a few settings for ZigBee device that are related to the radio core, UART and wireless network topology. They are presented in Table 4.

**Table 4.** Key settings for ZigBee device.

| Name | Value | SFR [1] or Function |
|---|---|---|
| Radio Channel | 2405 MHz | FREQCTRL.FREQ[6:0] |
| Receiver Sensitivity | −85 dBm | CCACTRL0[7:0] |
| Transmit Power | 4.5 dBm | TXPOWER[7:0] |
| Network Topology | Star | GenericApp_Init |
| Light AD Channel | AIN6, P0_6 | ADCCON3[7:0] |
| Gas AD Channel | AIN7, P0_7 | ADCCON3[7:0] |
| UART Selection | 0 | P0SEL, U0CSR |
| UART Baud Rate | 9600 | U0GCR, U0BAUD |

[1] Special Function Register.

IEEE 802.15.4, the basis of ZigBee protocol [45], specifies 16 channels within 2.4 HGz band in which the carrier frequency ranges from 2405 MHz to 2480 MHz with 5 MHz apart, and it is controlled by the 7-bit SFR FREQCTRL FREQ. The radio channel used in this project is configured as channel 11, while the carrier frequency is 2405 MHz. In addition, the receiver sensitivity is set to −85 dBm by the SFR CCACTRL0, while the transmit power is initialized as 4.5 dBm in SFR TXPOWER. For network topology in this project, star network is chosen for simplicity. The coordinator is responsible for establishing the ZigBee wireless network, processing the incoming join requests and managing the joined end devices in the network. The ZigBee end devices are equipped with multiple sensors for data collecting. Specifically, the light and gas sensor are connected with the Port 0_6 and Port 0_7 of CC2530, which means that the AD channel should be configured as AIN6 and AIN7 respectively in SFR ADCCON3 during the AD process. The UART communication between MT3620 and CC2530 ZigBee Coordinator is configured as 9600 baud, 8 data bits, 1 stop bit, no parity bit and no flow control. The UART configuration can be achieved by setting SFR P0SEL, U0CSR, U0GCR and U0BAUD.

### 4.2. Software

In this subsection, the necessary prerequisites, the software implementation for the device layer, including MT3620 device and ZigBee wireless sensor network device, are presented in detail. In addition, the source code and the corresponding tutorials are hosted on GitHub repository, please refer to the section of supplementary materials for more details.

### 4.2.1. Prerequisites

As shown in Table 5, a Windows 10 PC with IAR Embedded Workbench for 8051 and Visual Studio 2017 Community (Enterprise and Professional version are also supported) are leveraged to develop applications for CC2530 ZigBee nodes, MT3260 development kits and Windows desktop respectively. Azure Sphere software development kit (SDK) Version 18.11 [46] and Microsoft NET Framework Version 4.7.0 are also installed as indispensable extensions for Visual Studio 2017 Community. In order to establish connection with Azure IoT Hub, Microsoft.Azure.Devices and WindowsAzure.ServiceBus NuGet packages are included in the windows-form based project.

**Table 5.** Prerequisites for software design in the project.

| Target | Integrated Development Environment | Version | Language |
|--------|-----------------------------------|---------|----------|
| CC2530 | IAR Embedded Workbench for 8051 | 8.10 | C |
| MT3620 | Visual Studio 2017 Community | 15.9.5 | C |
| Desktop | Visual Studio 2017 Community | 15.9.5 | C# |

### 4.2.2. Software for MT3620 as Direct Connected Device

The application that runs in MT3620 as direct connected device is an executable program which is responsible for collecting the data of the sensors via onboard interfaces and transmitting the data to the Azure IoT Hub. Benefiting from the hardware support of Seeed Studio, MT3620 Grove Shield, Grove Temperature & Humidity Sensor (SHT31), Grove Sound Sensor, Grove Relay, Grove Light Sensor are connected to the MT3620 board. The prototype of this scenario is shown in Figure 4b.

The Wi-Fi network is connected and the necessary hardware interfaces are opened during the initialization period. In addition, Azure IoT related functions and callbacks including Azure IoT initialization, device twin update callback, connection status callback, message receive callback and direct method callback are executed in an ordered sequence. In the main loop, the data of the connected sensors are read via hardware interfaces periodically. The reading interval is set to be 30 s by default, which can be changed dynamically by direct method callback while the program is running. As soon as the data of sensors are obtained, the corresponding JSON values are set and serialized by JSON lib. Finally, at the end of the referred process, the message is transmitted to Azure IoT Hub.

Relying on the capability of Azure IoT Hub to invoke direct methods on the devices from the cloud, the DirectMethod callback function is registered in the initialization process and can be called as soon as a direct method call is received from the Azure IoT Hub. In the callback function, the time interval of sensor data collecting, as well as the time interval of device-to-cloud message sending are updated according to the JSON values that are set in the payload of the message. The response is encapsulated immediately on the message payload and sent as a reply to the previous callback function.

### 4.2.3. Software for MT3620 as Gateway

The MT3620 can also be configured as a gateway to allow legacy IoT systems to connect to the Internet and to the Azure IoT Hub. The UART interface of the Azure Sphere device is used to communicate with the ZigBee wireless sensor network, i.e., the legacy IoT system. Meanwhile, the UART peripheral should be included in the application manifest file, in which all the resources that the application requires are listed. Figure 5 shows the application manifest file of the project. It is noted that the application manifest will be accessed by Azure Sphere runtime to determine which capabilities are allowed to use as soon as the application is sideloaded or deployed to the device. Any attempt to access resources that are not described in the manifest will be denied by the runtime. Hence, the UART and Allowed Connections to Azure cloud services are added as essential capabilities.

The flowchart of the application that is running in the MT3620 as gateway is depicted in Figure 6. Basically, the initialization process for the application is pretty much the same as that in the MT3620 as direct connected device, except that the UART peripheral is configured and opened for communication with ZigBee Coordinator. In the main loop, the UART event handler is called for inbound message receiving, data validity checking and local parameters updating. As soon as the transmission timer expires, the latest parameters of the ZigBee wireless sensor network are sent to the Azure IoT Hub via JSON data.

```
{
   "SchemaVersion": 1,
   "Name": "AzureSphereAzureIoTHub",
   "ComponentId": "1977c307-e5d8-48f4-b268-aa754591b6bd",
   "EntryPoint": "/bin/app",
   "CmdArgs": [],
   "Capabilities": {
      "AllowedConnections": [ "global.azure-devices-provisioning.net", "MyIoTHubSample. azure-devices.net"],
      "AllowedTcpServerPorts": [],
      "AllowedUdpServerPorts": [],
      "Gpio": [8, 9, 10, 15, 16, 17, 18, 19, 20, 12, 13],
      "Uart": [ "ISU0" ],
      "WifiConfig": true,
      "NetworkConfig": false,
      "SystemTime": false,
      "DeviceAuthentication": "c2bc2beb-7ef0-47a3-b6d2-645e5d751926"
   }
}
```

**Figure 5.** The application manifest file for the project of MT3620 as gateway.

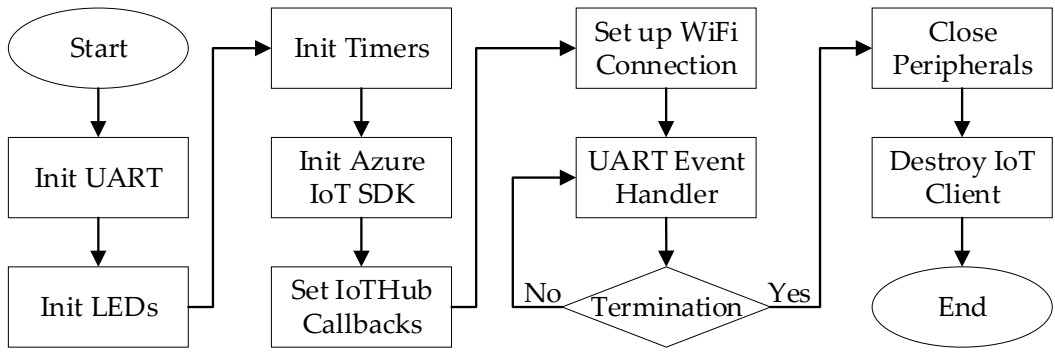

**Figure 6.** The flowchart of the application.

### 4.2.4. Software for ZigBee Wireless Sensor Network

There are four official protocol options for end users to choose from Texas Instruments for CC2530, i.e., SimpliciTI Compliant Protocol Stack, RF4CE Compliant Protocol Stack, IEEE802.15.4 Medium Access control (MAC) software stack and Z-Stack [47]. In this article, Z-Stack is used for wireless sensor nodes management and data transmission.

The program running in the CC2530 ZigBee node with Z-Stack executes as follows. After the initialization process, the Operating System Layer executes the main loop in which the task list will be accessed and checked in order. As soon as the condition is met, the specified task will be carried out immediately. For ZigBee Coordinators in this article, there are two main tasks, which are UART message handler task and ZigBee wireless network incoming package handler task. On the one hand, the UART message handler task deals with the incoming data from MT3620 and then relays the

message to ZigBee End Device via Z-Stack protocol message. On the other hand, the wireless network incoming package handler task is responsible for processing with the Z-Stack protocol message from ZigBee End Device and transmitting the valid data to MT3620 via UART interface. For ZigBee End Device equipped with different kind of sensors, there are two main tasks as well. Firstly, reading data from sensors and transmitting message to ZigBee Coordinator are completed in the Data Send Handler event within a user defined interval. Secondly, the downlink messages from ZigBee Coordinator through Z-Stack protocol are handled in the Message Process event.

The payload of the uplink message is shown in Figure 7. The payload is prefixed with one-byte Sensor ID, which is closely followed by two-byte temperature data, two-byte humidity data, two-byte light data, two-byte gas data and one-byte PIR sensor data.

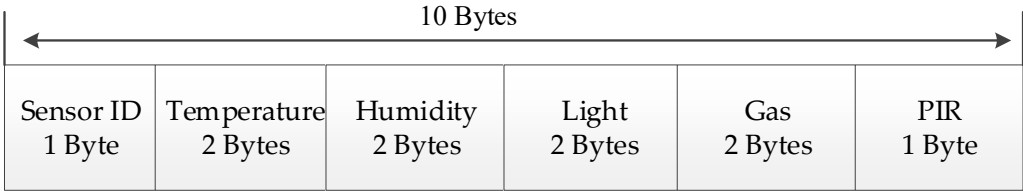

**Figure 7.** The payload of the uplink message.

### 4.3. Azure Cloud Services

Getting access to Azure Table storage is not only fast but also cost-effective for user applications. The entities of the table are shown in Table 6, which contain multiple properties including partition key, row key, time stamp, device ID, event enqueued UTC time, event processed UTC time, gas, humidity, light, sound and temperature. The partition key, row key and time stamp are automatically generated for every entity in the table, which are representative of the first part of an entity's primary key, the second part of an entity's primary key and the time that the entity was last modified.

**Table 6.** Azure Storage Table.

| Properties | Type |
|---|---|
| Partition key | String |
| Row key | String |
| Timestamp | DateTime |
| Device ID | String |
| Enqueued UTC time | DateTime |
| Processed UTC time | DateTime |
| Gas | Analogue |
| Humidity | Int64 |
| Light | Int64 |
| Sound | Int64 |
| Temperature | Int64 |

Power BI [48] is selected as data visualization tool to show the insights of the sensor data. As soon as the Azure Stream Analytics starts running, the incoming messages with the JSON format will be streamed to Power BI online service. Furthermore, the dataset becomes available in user's workspace for reports editing and generation. In order to fit the visual reports to different platforms, the Web Layout and Mobile Layout are both provided. Figure 8 shows the reports in editor view including average temperature, average humidity, maximum light, minimum sound, real-time sound and light, real-time temperature and humidity.

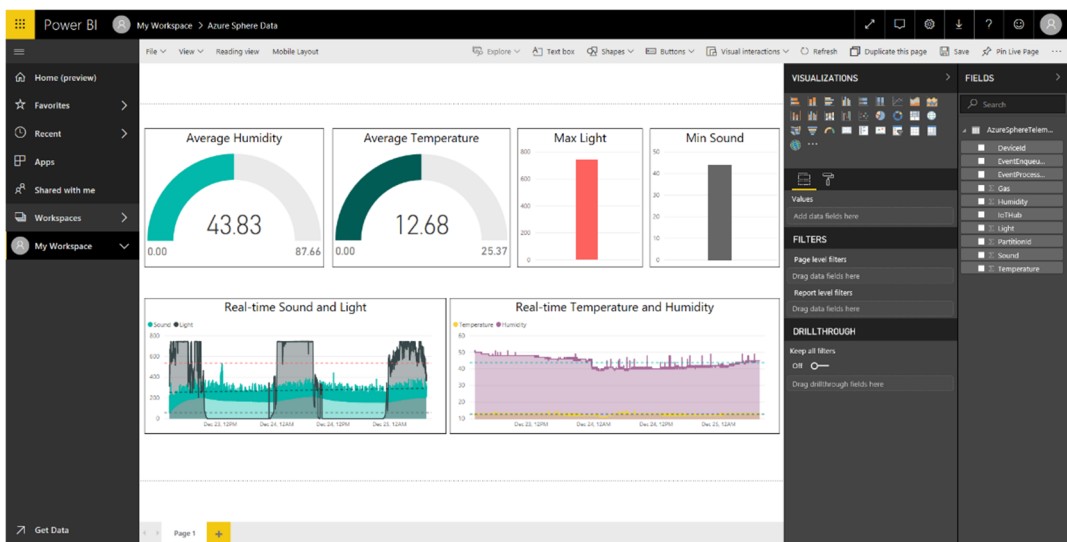

**Figure 8.** Editor view for Power BI reports.

## 5. Experimental Results

The proposed IoT integrated solution that includes two user scenarios, in which the Azure Sphere device is designed to act as a direct connected device or a gateway for legacy nodes, is designed and developed. In this section, the testbed is demonstrated in the views of device layer, cloud layer and application layer.

### 5.1. Device Layer

The device layer is mainly composed of MT3620 Azure Sphere Kit, ZigBee wireless sensor network devices and different types of sensors. First, data collected by ZigBee end device will be transmitted to ZigBee Coordinator via WSN. The data can be captured and parsed frame by frame with SmartRF Packet Sniffer, which is shown in Figure 9. It is clearly depicted that not only the original data, but also the protocol details of the frame are unveiled in detail. For example, the highlighted frame in Figure 9 indicates that it is a data request message from the ZigBee coordinator. The length of the frame, the details of the frame control field, the destination address, the source address, the link quality indication and the frame check sequence are also parsed.

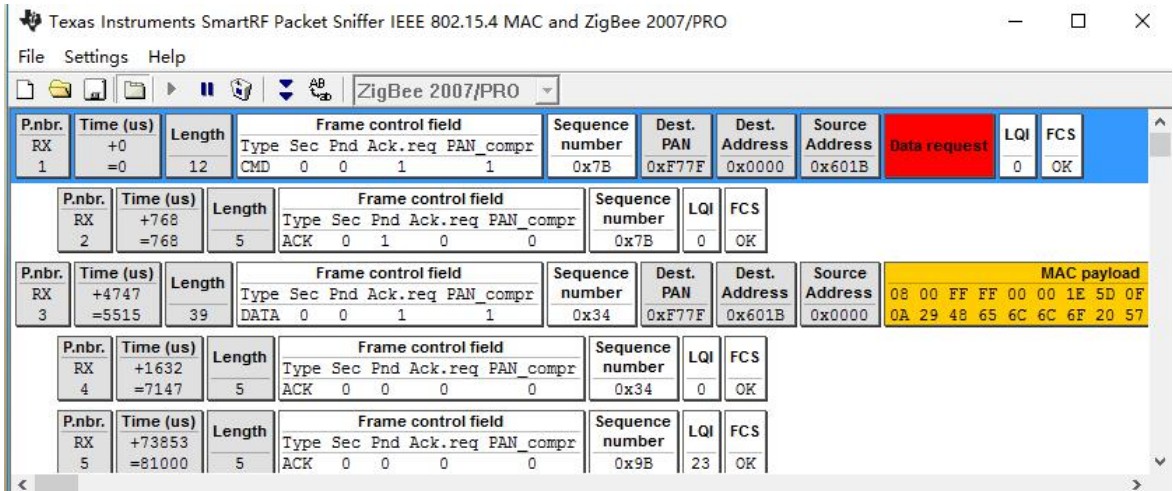

**Figure 9.** ZigBee data captured by Packet Sniffer.

Figure 10 shows the captured waveforms for UART communication between the MT3620 gateway and CC2530 coordinator device, which are generated by Tektronix digital oscilloscope TBS1064. The sending and receiving UART waveforms are demonstrated in Figure 10a,b respectively. It is noted that the UART serial communication here is at TTL (Transistor-Transistor Logic) level, which means that the voltage will always remain between 0V and Vcc.

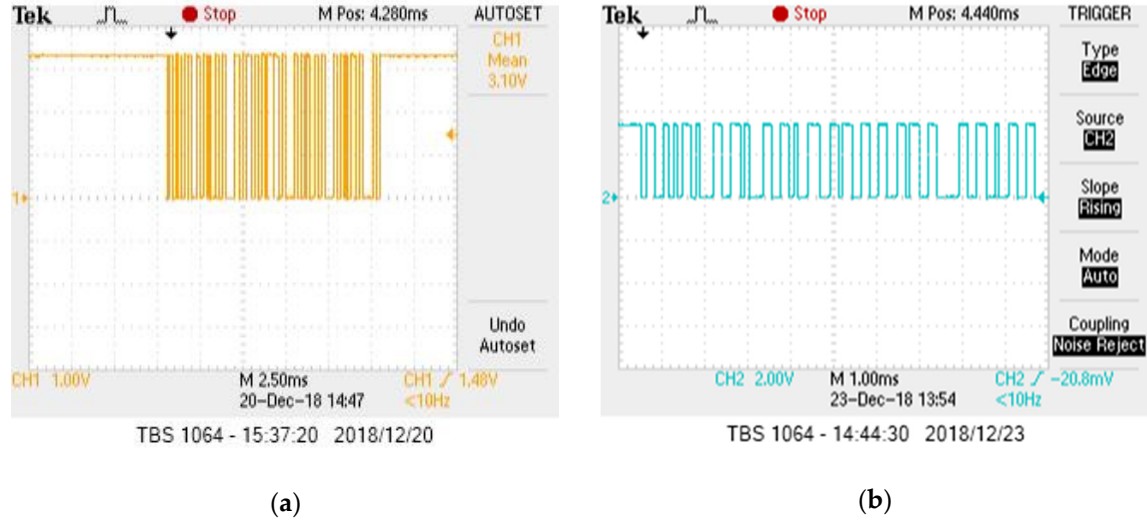

(**a**)                                                     (**b**)

**Figure 10.** UART Waveforms: (**a**) Transmit; (**b**) Receive.

*5.2. Cloud Layer*

In the cloud layer, all the device-to-cloud and cloud-to-device messages via Azure IoT Hub can be monitored by Device Explorer Twin. As shown in Figure 11, the data and time of the message, device name, device ID as well as sensor data are displayed in order within JSON format.

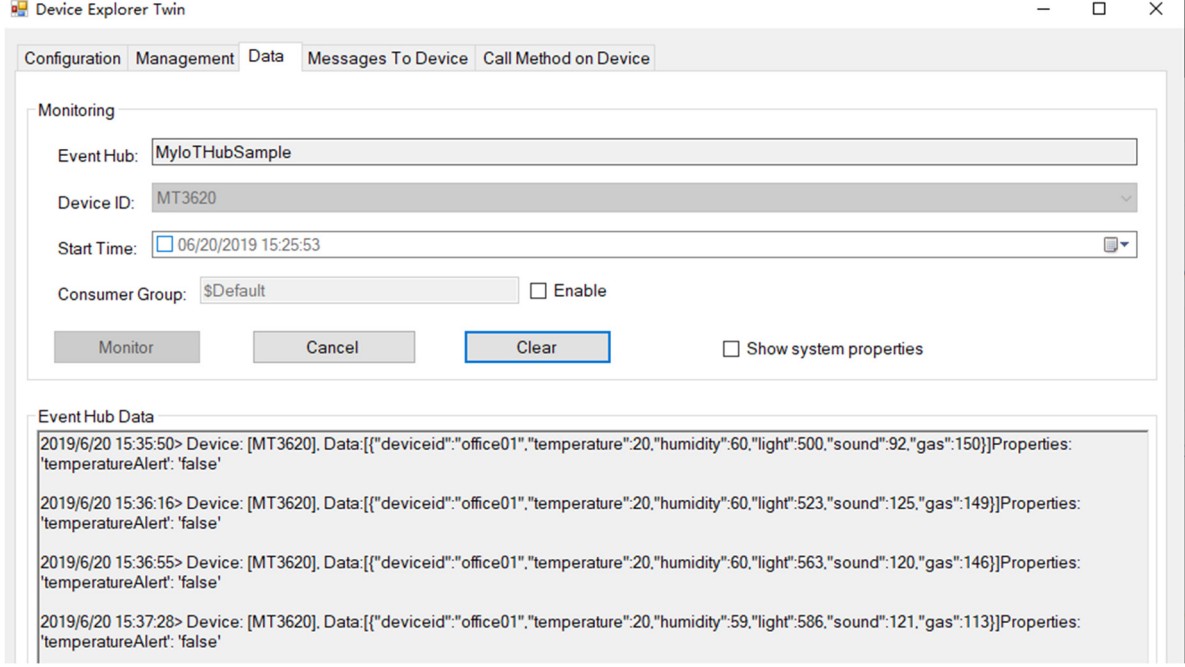

**Figure 11.** Azure IoT Hub Data Monitor.

When the JSON packages arrive at the Azure IoT Hub, they will be processed in order. It takes time for the Azure IoT Hub to retrieve the messages and stream them to the Azure Stream Analytics. In order to evaluate the latency (milliseconds), all the messages from 2:00 am to 8:00 pm, 28 June 2019, are exported and analyzed. By the properties of "EventEnqueuedUtcTime" and "EventProcessedUtcTime", the latency of every message in Azure IoT Hub can be calculated. The average latency, the maximum latency and the minimum latency are depicted in Figure 12.

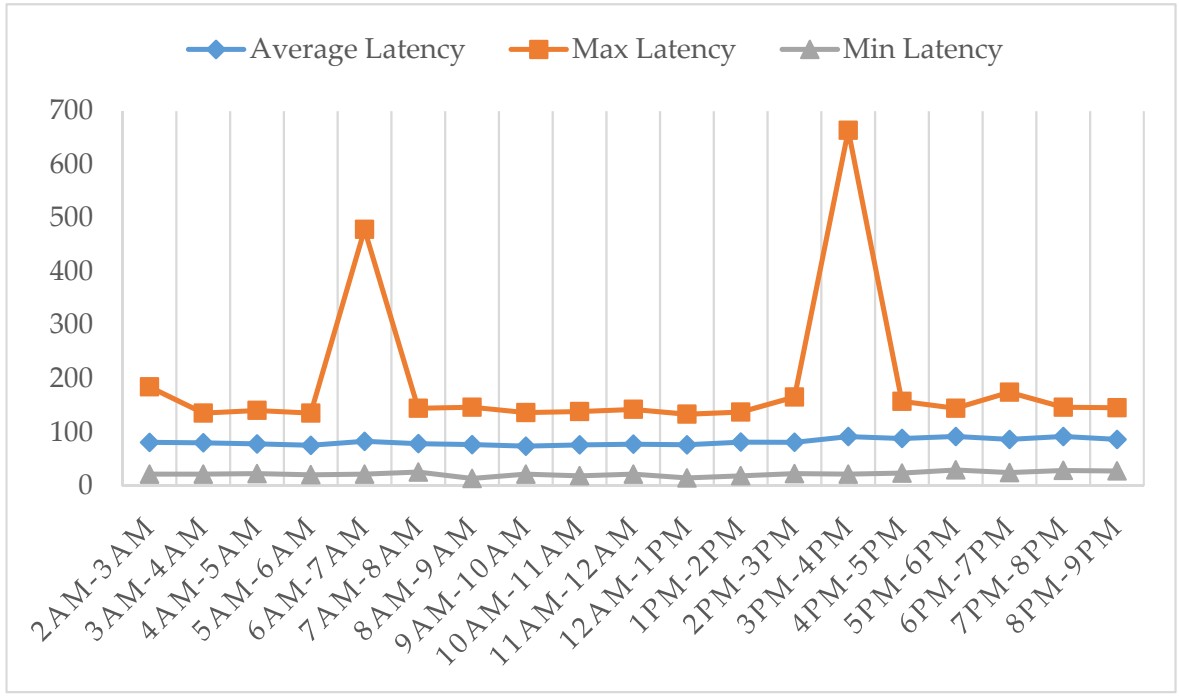

**Figure 12.** Latency evaluation for Azure IoT Hub.

According to the statistics in Figure 12, the minimum latency and the average latency are round ten milliseconds and hundred milliseconds, respectively. The maximum latency can reach 500 milliseconds and 650 milliseconds in 6:00 am–7:00 am and 3:00 pm–4:00 pm. Though they are relatively large compared with average latency, it is still acceptable for most of time-sensitive IoT applications.

After real-time filtering by Azure Stream Analytics, the data is sent to Azure Table Storage. Azure Table Storage is used for saving data from sensors. The detail information of the table "Azure Sphere MT3620" is shown by Microsoft Azure Storage Explorer in Figure 13.

In order to evaluate the performance of Azure Table Storage, the average end-to-end latency (AverageE2ELatency) of successful requests and the average server latency (AverageServerLatency) are provided by Azure. AverageE2ELatency includes the required processing time within Azure Table Storage to read the request, send the response and receive acknowledgement of the response. AverageServerLatency stands for the average latency used by Azure Table Storage to process a request, excluding failed requests. This value does not include the network latency which is included in AverageE2ELatency. Latency evaluation for Azure Table Storage is shown in Figure 14.

According to the curves in Figure 14, the values of AverageServerLatency and AverageE2ELatency from June 28 to June 29 are 11.25 ms and 77.27 ms respectively, which means that the average entire latency is less than 100 ms. It is acceptable for most of IoT applications.

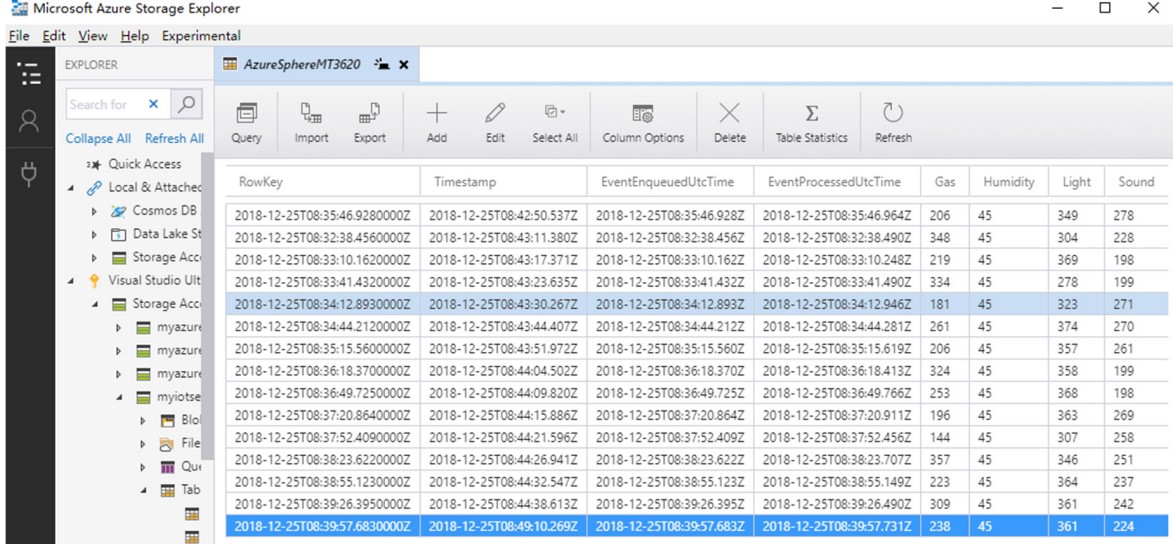

**Figure 13.** Dataset in Azure Table Storage.

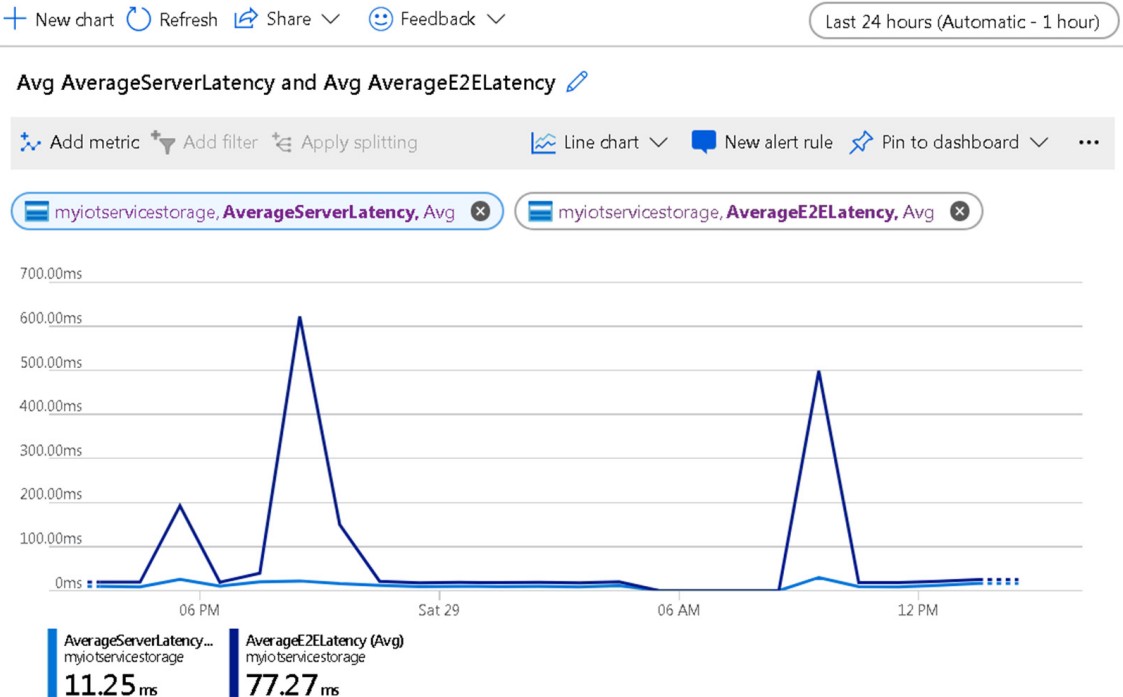

**Figure 14.** Latency evaluation for Azure Table Storage.

## 5.3. Application Layer

In the application layer, a windows-form based native application is developed for remote monitoring with Visual Studio 2017 Community IDE. In addition, the Power BI Desktop and Power BI Mobile can be downloaded in Microsoft Store, Apple App Store and Google Play for Windows, iOS and Android devices respectively.

As shown in Figure 15, all the messages that are sent from MT3620 devices to Azure IoT Hub are obtained, parsed and displayed by the native application for real-time monitoring. Meanwhile, the threshold can be set by the user. As soon as the value exceeds the threshold, the cloud-to-device message will be sent by the application immediately to drive the remote relay in time.

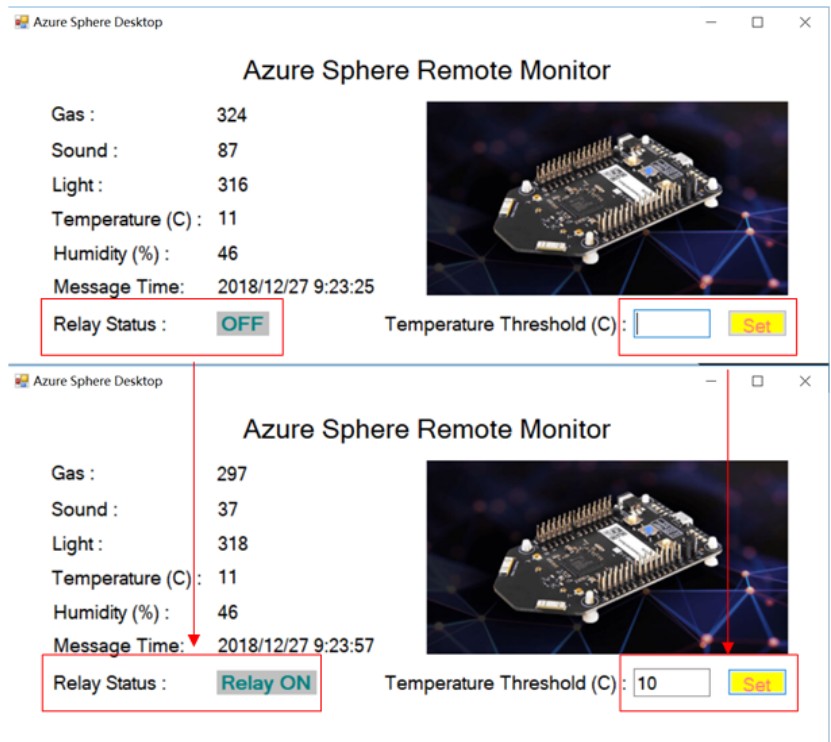

**Figure 15.** The layout of windows-form based native application.

For data visualization, Figure 16 presents the screens of the applications that show the designed reports on both Power BI for Desktop and Power BI for Android.

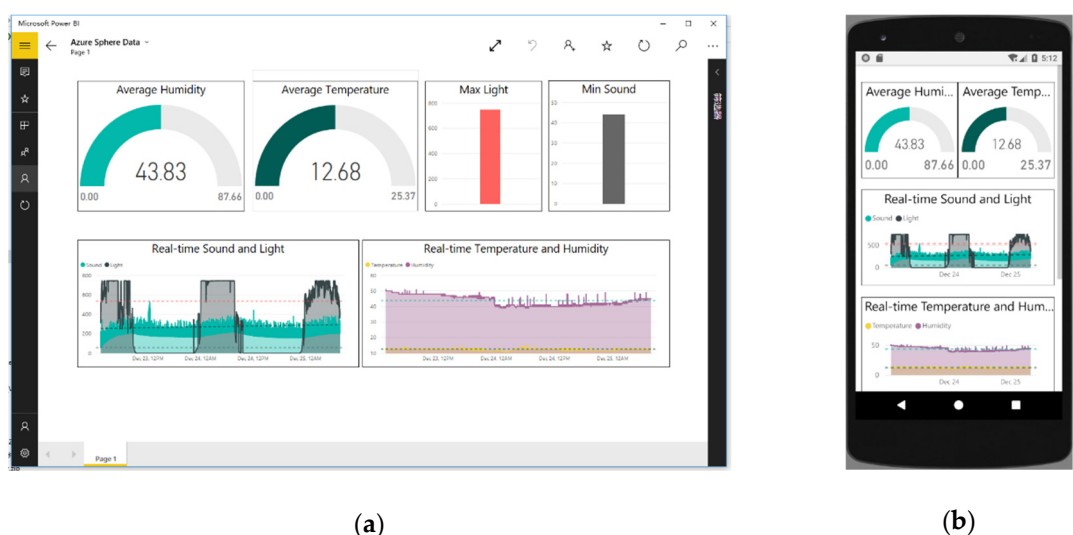

(**a**)                                               (**b**)

**Figure 16.** Designed reports on different platforms: (**a**) Windows Desktop; (**b**) Android device.

## 6. Discussion

In this article, we investigated the integration for IoT solutions with large-scale, low-cost and secure end devices. We focused on the following three aspects. Firstly, featured by Azure Sphere device and Azure cloud services, a novel integration of hardware, software and services was designed, deployed and tested. Secondly, the hardware prototypes including MT3620 direct connect device and MT3620 gateway device were designed and implemented. Thirdly, the programs for hardware devices,

the configurations for cloud services as well as the applications for Windows devices were designed and tested.

According to the research in [12–17], a lot of end devices in industry, healthcare and other general user scenarios mostly consist of sensors and actuators, making them susceptible to threats and attacks due to the missing of security and privacy design for those low-cost IoT devices. Furthermore, most of the current solutions have considered the security threats and strategies in one specific layer or multiple layers. However, very few of them have treated security all the way from the device to the cloud. To address these issues, Azure Sphere certified MT3620 is used to serve as direct connected device and gateway device with low cost and high security in the proposed solution. Meanwhile, the security policies and rules are designed to work together in a harmonious whole from device, connection and cloud perspectives. Furthermore, profiting from Azure IoT Hub, Azure Device Provisioning Service and Azure Sphere Security Service, the lager-scale device provision, configuration and management can be achieved in a secure and efficient way. In summary, this article is concentrated on the integration of hardware, software and services that is based on Azure Sphere device and Azure cloud services to achieve security all the way from every low-cost device to the cloud simultaneously. As such, one limitation of the work is that the integration of the available devices and services for large-scale scenarios with limited budgets, rather than the creation of security policies and rules is concentrated upon. Another limitation of current work is that the system response to the real security threats is not considered yet. However, according to the literature [8,49], with the Azure IoT platform, which is built with high security and privacy, along with Security development lifecycle (SDL) [50] and Operational Security Assurance (OSA) [51] for secure development and operation, the overall security can be achieved.

## 7. Conclusions

As our society embraces the IoT and cloud technologies in a wide range of application areas including industry, agriculture and education, considerations for security and privacy must be more than an afterthought. In this sense, the focus needs to shift from securing network perimeters to safeguarding data spread across devices, networks and the cloud as a whole system. To this end, an Azure Sphere based integrated solution for IoT, which utilizes the highly secure, low-cost MCU MT3620 and Microsoft Azure cloud services to meet the requirements of robust end-to-end security and unified device management user scenarios, is proposed in this article. The overall system has been completely designed from hardware prototype to software implementation, and from the device to the cloud. With the proposed solution, users are capable of integrating the highest levels of security into their products and services to address the rising risks and threats. Furthermore, device provision, configuration and management are implemented by Azure cloud services. In order to test and verify the proposed system, an experimental testbed is established on the basis of MT3260 development kit, CC2530 ZigBee wireless network, Azure cloud services, Power BI and native windows-form applications. The experimental results are demonstrated from the device layer, the cloud layer and the application layer respectively, indicating that the proposed integrated solution is feasible and effective.

**Supplementary Materials:** The source code and guidelines of this project for MT3620, CC2530 and windows-form based native applications are hosted on GitHub repository: https://github.com/shijiong/A-Security-Solution-Using-Azure-Sphere-for-IoT.

**Author Contributions:** Conceptualization, J.S.; Methodology, L.J.; hardware, J.S.; software, J.S. and L.J.; validation, J.S.; investigation, L.J.; resources, J.S.; data curation, L.J.; writing—original draft preparation, J.S. and J.L.; writing—review and editing, J.L.; project administration, J.S.

**Funding:** This research was supported by the Zhejiang Provincial Public Welfare Program under Grant (No. 2017C33152 and No. 2017C31040), Scientific Research Fund of Zhejiang Provincial Education Department under Grant (No. Y201636753), Research Development Fund of Zhejiang Wanli University, Ningbo Technology Project for People's Welfare (No. 2017C50028).

**Acknowledgments:** The authors would like to thank all the anonymous reviewers for their insightful comments and suggestions.

**Conflicts of Interest:** The authors declare no conflict of interest.

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
