# Peer review of "The Integration of Azure Sphere and Azure Cloud Services for Internet of Things"

_applsci, doi:10.3390/app9132746_

Round 1

Reviewer 1 Report

General Comments:

When we start reading the paper, it firstly looks dedicated to the implementation of a fully integrated solution capable of ensuring security on an IoT platform, from the devices level to the cloud level. Nevertheless, the security aspect disappears along with the paper. The authors, in section 5, concluded that the work lacks of security tests evaluation. Thus, this counterposes the introduced on first sections. In my opinion, if the work is not dedicated to security it shall be clear after the introduction that the work is focused only in the integration of an IoT platform, trying to justify with knowledge gathered from the literature that the security would be ensured.

Now, revisiting all paper:

0.      Abstract: In my opinion it shall be enhanced. We can understand that the authors propose the implementation of user cases of tools of Azure, but by reading the abstract it is not perfectly clear what is the goal, why the work is important and what are the main outcomes of the works. It shall be clear after reading the abstract, what was the problem the authors wanted to solve, how they planned to do it and what were the results (in a summary way).

1.      Introduction:

a.      The sentence of lines 34,35 and 36 lacks of reference;

b.      The second paragraph is, in my opinion, confusing. It could be more objective and well structured;

c.      The third paragraph is the state of the art on security aspects? I think it is weak. The authors shall notice that the state of the art is merely an enumeration with small details. The reader is not capable of connecting the present work with previous works. Then, in my opinion, there is a worse issue with this state of the art. As stated in the general comments, the work is particularly focused on the integration of a fully IoT platform. So, maybe the state of the art shall cover also fully integrated IoT platforms. Why this platform is better than others? Or there arent other platforms? Also, the work is totally focused on Azure Concerning this, there arent any previous works with Azure? Even if with different tools?

d.      Section 1.1 is lost in the introduction. I would take it out. Azure can be introduced on section 2 for example. Of course the authors shall introduce Azure on the introduction but objectively and shortly. Also, Figure 1 is weak. It does not bring any special information.

e.      I like subsection 2.2. Only here the reader can understand that the work is focused in the implementation of particular user cases. However, I think it lacks of some explanation. Why these user cases were defined and how they can be used to take more generic conclusions?

2.      System overview

a.      The 2nd paragraph is not clear. Shall be re-written

b.      In my opinion, an introduction of Azure and its modules would enrich the paper and help the reader to understand better the system. Additionally, in Figure 2, all the modules that have a direct connection to Azure modules shall be represented.

c.      Lines 160,161, and 162: is this the only way of categorising the security rules and policies? This is stated by authors or it is defended by the literature? Table 1 is a summary of security specification of Azure? When we start reading the paragraph it looks more generic but after inspecting the table it looks like specific from Azure. Please make this clear.

3.      System Design and implementation

a.      On line 131 the authors already spoke about a system architecture. Here it is presented another. It is important to clarify what is introduced in one and another.

b.      Figure 3 has bad quality and if I am not wrong I am not able to find any explanation;

c.      Lines 185 and 186: authors state that Azure cloud services are not mandatory and that another cloud can be used. But for curiosity, from which point of the system exists interoperability? What would need another cloud platform to support the operation with MT3620?

d.      The firsts paragraph of subsection 3.1. could be better written. The scenarios are not clear.

e.      Subsection 3.2: Azure sphere MT3620 is more and less explained. The same does not happen with Azure IoT Hub. Maybe the subsection explaining all modules of Azure would help;

f.       Figure 5: what is the Azure Sphere Security Service? It is some module inside the Azure Sphere Device? Without clarifying this I am not able to understand Figure 5.

g.      Subsection 3.3.2 Maybe a flowchart would be helpful.

h.      Subsection 3.3.4 Why authors choose Z-stack?

i.       Subsection 3.4: Azure IoT Hub. Only here we understand what is the role of the Hub. But it shall be clarified earlier. It is not clear how the data is treated. How does the cloud know the messages that are going to receive? Where is that configured/programmed? This part does not look at an integration of services. It looks more a description of Azure Cloud Services

j.       I am not sure about the suitability of the organization of the paper regarding section 3 and 4.. System design and implementation in the same section usually is not a good idea. The design is one thing the Implementation is another. If the implementation is small maybe it can be integrated at the beginning of the results. The system design shall be something generic. The implementation is a specific implementation of such design (where you choose the components, the platforms etc).

4.      Results:

a.      Figure 9 is unnecessary. If I see well it only describes the operation of a UART. That is well known already. Authors dont need to show it;

b.      Table 4: these are configurations that shall appear on implementation. Thus, my comment 3.j.) continues to make sense; Subsection 4.1. are not results;

c.      Also in 4.2 and 4.3 I have difficulties in seeing that as results. Results are measurements that show that the platform works well i.e. that the messages sent by the MT 3620 are propagated until the cloud with a certain latency, throughput, etc. Do the authors have something that proves the correct operation of the system?

5.      Discussion: maybe this table would fit on the state of the art if well detailed. In fact, the lack of security test is a big limitation of the work. Even if there arent tests, couldnt the authors provide some comments about what they expect (for example using the literature as support?)

There are some misspellings but I would prefer to state them after the major review. 

Author Response

Response to Reviewer 1 Comments

We thank the reviewers for their constructive comments, which have helped improve the quality of the manuscript. The comments are well taken and the manuscript has been revised accordingly. In the following, we provide an itemized response to the comments and questions raised by the reviewers. For the reviewers’ convenience, their original comments are copied here and shown in italics, and our responses are highlighted in red.

General Comments:

When we start reading the paper, it firstly looks dedicated to the implementation of a fully integrated solution capable of ensuring security on an IoT platform, from the devices level to the cloud level. Nevertheless, the security aspect disappears along with the paper. The authors, in section 5, concluded that the work lacks of security tests evaluation. Thus, this counterposes the introduced on first sections. In my opinion, if the work is not dedicated to security it shall be clear after the introduction that the work is focused only in the integration of an IoT platform, trying to justify with knowledge gathered from the literature that the security would be ensured.

Response to General Comments: We greatly appreciate the honourable reviewer for the constructive suggestion. In the revision, we modify the title of this article from “The Integration of Azure Sphere and Azure Cloud Services for Security-Oriented Internet of Things” to “The Integration of Azure Sphere and Azure Cloud Services for Internet of Things”. We make clear that this article is mainly focused on the integration solution for IoT application. And then we justify with knowledge and practice that the high security, low cost and easy device management would be ensured via proposed integration.

Point 1: Abstract: In my opinion it shall be enhanced. We can understand that the authors propose the implementation of user cases of tools of Azure, but by reading the abstract it is not perfectly clear what is the goal, why the work is important and what are the main outcomes of the works. It shall be clear after reading the abstract, what was the problem the authors wanted to solve, how they planned to do it and what were the results (in a summary way).

Response 1: We greatly appreciate the honourable reviewer for the detailed suggestion. In the revision, we rewrite the abstract. First, we state the current situation and main issues in the IoT integration solutions. Then we propose our solution that based on Azure Sphere and Azure cloud services. And finally, the results are concluded in a summary way.

Point 2:

1.      Introduction:

a.      The sentence of lines 34,35 and 36 lacks of reference;

Response 2: We greatly appreciate the honourable reviewer for the detailed suggestion. In the revision, we add the reference in line 79.

Point 3:

b.      The second paragraph is, in my opinion, confusing. It could be more objective and well structured;

Response 3: We greatly appreciate the honourable reviewer for the detailed suggestion. In the revision, we rewrite the introduction and related wok. This paragraph is integrated into Subsection 2.1 Security Research in IoT Systems.

Point 4:

c.      The third paragraph is the state of the art on security aspects? I think it is weak. The authors shall notice that the state of the art is merely an enumeration with small details. The reader is not capable of connecting the present work with previous works. Then, in my opinion, there is a worse issue with this state of the art. As stated in the general comments, the work is particularly focused on the integration of a fully IoT platform. So, maybe the state of the art shall cover also fully integrated IoT platforms. Why this platform is better than others? Or there aren’t other platforms? Also, the work is totally focused on Azure… Concerning this, there aren’t any previous works with Azure? Even if with different tools?

Response 4: We greatly appreciate the honourable reviewer for the detailed suggestion. In the revision, we rewrite the introduction and related wok. The state of the art is located in Section 2. It is comprised of Subsection 2.1 Security Research in IoT Systems and Subsection 2.2 Azure Sphere. In Subsection 2.1, the security in IoT integration solutions are reviewed according to the different IoT layers. And the details are compared in Table 1.

Point 5:

d.      Section 1.1 is lost in the introduction. I would take it out. Azure can be introduced on section 2 for example. Of course the authors shall introduce Azure on the introduction but objectively and shortly. Also, Figure 1 is weak. It does not bring any special information.

Response 5: We greatly appreciate the honourable reviewer for the constructive suggestion. In the revision, we rewrite the introduction and related wok. The Azure Sphere is located in Subsection 2.2. Furthermore, Figure 1 is deleted in the revision.

Point 6:

e.      I like subsection 2.2. Only here the reader can understand that the work is focused in the implementation of particular user cases. However, I think it lacks of some explanation. Why these user cases were defined and how they can be used to take more generic conclusions?

Response 6: We greatly appreciate the honourable reviewer for the detailed suggestion. In the revision, we move system security to Subsection 3.5. Furthermore, we add application security in the revision.  The reference 41 are added in line 281 to explain why we consider the system security as device security, connection security, cloud security and application security.

Point 7:

2.      System overview

a.      The 2nd paragraph is not clear. Shall be re-written

Response 7: We greatly appreciate the honourable reviewer for the detailed suggestion. In the revision, we rewrite this paragraph and it is located from line 165 to line 183 in the Subsection 3.1.

Point 8:

b.      In my opinion, an introduction of Azure and its modules would enrich the paper and help the reader to understand better the system. Additionally, in Figure 2, all the modules that have a direct connection to Azure modules shall be represented.

Response 8: We greatly appreciate the honourable reviewer for the constructive suggestion. In the revision, we rewrite the overview of the proposed integration and it is located from line 160 to line 202 in the Subsection 3.1. All modules are explained in detail.

Point 9:

c.      Lines 160,161, and 162: is this the only way of categorising the security rules and policies? This is stated by authors or it is defended by the literature? Table 1 is a summary of security specification of Azure? When we start reading the paragraph it looks more generic but after inspecting the table it looks like specific from Azure. Please make this clear.

Response 9: We greatly appreciate the honourable reviewer for the detailed suggestion. In the revision, we move system security to Subsection 3.5. Furthermore, we add application security in the revision.  The reference 41 are added in line 281 to explain why we consider the system security as device security, connection security, cloud security and application security. The security considerations in Table 2 are concluded based on the Azure cloud services and device security information.

Point 10:

3.      System Design and implementation

a.      On line 131 the authors already spoke about a system architecture. Here it is presented another. It is important to clarify what is introduced in one and another.

Response 10: We greatly appreciate the honourable reviewer for the detailed suggestion. Actually, here we state the proof of concept for the proposed integration. In the revision, we rewrite the system implementation in Section 4. The connections between the two figures are explained from line 318 to line 325.

Point 11:

b.      Figure 3 has bad quality and if I am not wrong I am not able to find any explanation;

Response 11: We greatly appreciate the honourable reviewer for the detailed suggestion. In the revision, we move it to the Section 4. And it is explained from line 318 to line 325.

Point 12:

c.      Lines 185 and 186: authors state that Azure cloud services are not mandatory and that another cloud can be used. But for curiosity, from which point of the system exists interoperability? What would need another cloud platform to support the operation with MT3620?

Response 12: We greatly appreciate the honourable reviewer for the detailed suggestion. Yes, according to the current samples provided by Microsoft, we cannot find any solution other than Azure cloud. In the revision, we delete this sentence.

Point 13:

d.      The firsts paragraph of subsection 3.1. could be better written. The scenarios are not clear.

Response 13: We greatly appreciate the honourable reviewer for the detailed suggestion. In the revision, we move it to the Subsection 4.1. And it is rewritten from line 327 to line 332.

Point 14:

e.      Subsection 3.2: Azure sphere MT3620 is more and less explained. The same does not happen with Azure IoT Hub. Maybe the subsection explaining all modules of Azure would help.

Response 14: We greatly appreciate the honourable reviewer for the detailed suggestion. In the revision, Azure IoT Hub is first introduced in Subsection 3.3. It shall help the readers understand the process more easily.

Point 15:

f.       Figure 5: what is the Azure Sphere Security Service? It is some module inside the Azure Sphere Device? Without clarifying this I am not able to understand Figure 5.

Response 15: We greatly appreciate the honourable reviewer for the constructive suggestion. In the revision, Azure Sphere Security Service is first introduced from line 143 to line 145 in Subsection 2.2.

Point 16:

g.      Subsection 3.3.2 Maybe a flowchart would be helpful.

Response 16: We greatly appreciate the honourable reviewer for the detailed suggestion. Since the application that runs in the MT3620 as Direct Connect Device is quite the same as that as Gateway Device, and the flowchart is provided in Figure 6 in the revision, so the flowchart of the MT3620 as Direct Connect Device is not provided here.

Point 17:

h.      Subsection 3.3.4 Why authors choose Z-stack?

Response 17: We greatly appreciate the honourable reviewer for the detailed suggestion. Since Z-stack is provided by TI officially and is widely used in the ZigBee applications, so we choose Z-Stack in this article.

Point 18:

i.       Subsection 3.4: Azure IoT Hub. Only here we understand what is the role of the Hub. But it shall be clarified earlier. It is not clear how the data is treated. How does the cloud know the messages that are going to receive? Where is that configured/programmed? This part does not look at an integration of services. It looks more a description of Azure Cloud Services…

Response 18: We greatly appreciate the honourable reviewer for the detailed suggestion. In the revision, the integration of Azure IoT Hub is written in Subsection 3.3. And yes, Azure IoT Hub is a web service provided by Microsoft. We just configure this service on Azure Portal, and it does not need to program. And we also provide a figure (Figure 11 on page 15) to show the messages of Azure IoT Hub.

Point 19:

j.       I am not sure about the suitability of the organization of the paper regarding section 3 and 4.. System design and implementation in the same section usually is not a good idea. The design is one thing… the Implementation is another. If the implementation is small maybe it can be integrated at the beginning of the results. The system design shall be something generic. The implementation is a specific implementation of such design (where you choose the components, the platforms etc).

Response 19: We greatly appreciate the honourable reviewer for the detailed suggestion. In the revision, we divide the original Section 3 “System Design and Implementation” into two sections, i.e., Section 3 “System Design” and Section 4 “System Implementation”.

Point 20:

4.      Results:

a.      Figure 9 is unnecessary. If I see well it only describes the operation of a UART. That is well known already. Authors don’t need to show it;

Response 20: We greatly appreciate the honourable reviewer for the detailed suggestion. We think that the data flow is important and Figure 9 in the original version is a demonstration of UART data communication. Furthermore, we add Figure 9 and Figure 11 in the revision to show the ZigBee data captured by Packet Sniffer and Azure IoT Hub data captured by Device Explorer Twin. However, if the honourable reviewer thinks the figure of UART is useless, then we can delete it as well.

Point 21:

b.      Table 4: these are configurations that shall appear on implementation. Thus, my comment 3.j.) continues to make sense; Subsection 4.1. are not results;

Response 21: We greatly appreciate the honourable reviewer for the detailed suggestion. Yes, we agree with the honourable reviewer that most part of Subsection 4.1 are not results. Hence, we move them to Subsection 4.1 in the revision. Instead, we add Figure 9 in the revision to show the ZigBee data.

Point 22:

c.      Also in 4.2 and 4.3 I have difficulties in seeing that as results. Results are measurements that show that the platform works well i.e. that the messages sent by the MT 3620 are propagated until the cloud with a certain latency, throughput, etc. Do the authors have something that proves the correct operation of the system?

Response 22: We greatly appreciate the honourable reviewer for the detailed suggestion. In the revision, we add Figure 11 in the revision to show the Azure IoT Hub messages. Furthermore, we analyse the data in the Azure Table Storage right below Figure 12 to show that the latency is acceptable for most of time-sensitive IoT applications.

Point 23:

5.      Discussion: maybe this table would fit on the state of the art if well detailed. In fact, the lack of security test is a big limitation of the work. Even if there aren’t tests, couldn’t the authors provide some comments about what they expect (for example using the literature as support?)

Response 23: We greatly appreciate the honourable reviewer for the detailed suggestion. In the revision, we move the comparisons of the surveyed papers and the table to the Subsection 2.1. Furthermore, we refer to literature 8, 47, 49 and 50 from line 537 to 540 to support that the proposed integration is secure.

Reviewer 2 Report

* This article describes novel security solutions in Azure cloud including Azure Sphere device, Azure IoT Hub, Azure Sphere Security Service, Azure Device Provisioning Service, Azure Stream Analytics, Azure Table Storage and Power BI, …
* Authors also did an experimentation where  MT3620, the first certified Azure Sphere MCU is used to establish a testbed containing two different user scenarios in which MT3620 is utilised as direct connect device and gateway device respectively. 

* The overall comments:
    * The introduction section is too long. The related work and Azure Sphere Overview, presented in Introduction, should be the separate sections.
    * The overall presentation of the paper is good. However, I suggest that section 2 “System overview” should be improved by explaining in detail each module in Figure 2 (system overview) in the form of bullet point or numbering so that it is easy to read.
    * If I am not confused, Figure 3 is designed based on Figure 2. We should say a few words to link them.

Author Response

Response to Reviewer 2 Comments

We thank the reviewers for their constructive comments, which have helped improve the quality of the manuscript. The comments are well taken and the manuscript has been revised accordingly. In the following, we provide an itemized response to the comments and questions raised by the reviewers. For the reviewers’ convenience, their original comments are copied here and shown in italics, and our responses are highlighted in red.

Point 1: The introduction section is too long. The related work and Azure Sphere Overview, presented in Introduction, should be the separate sections.

Response 1: We greatly appreciate the honourable reviewer for the constructive suggestion. In the revision, we rewrite the instruction and related work. The related work and Azure Sphere overview are moved to Section 2 “Related Work”.

Point 2:

The overall presentation of the paper is good. However, I suggest that section 2 “System overview” should be improved by explaining in detail each module in Figure 2 (system overview) in the form of bullet point or numbering so that it is easy to read.

Response 2: We greatly appreciate the honourable reviewer for the detailed suggestion. In the revision, we divide the original Section 3 “System Design and Implementation” into two sections, i.e., Section 3 “System Design” and Section 4 “System Implementation”. And each module in Figure 1 (Figure 2 in the original manuscript) are explained in detail from line 165 to line 202 in the Subsection 3.1.

Point 3:

If I am not confused, Figure 3 is designed based on Figure 2. We should say a few words to link them.

Response 3: We greatly appreciate the honourable reviewer for the constructive suggestion. In the revision, we rewrite the system implementation in Section 4. Figure 3 (Figure 2 in the original manuscript) is the proof of concept for the proposed integration that depicted in Figure 1 (Figure 2 in the original manuscript). The connections between the two figures are explained from line 318 to line 325.

Reviewer 3 Report

In this paper, authors proposed a security integration including Azure Sphere device and 

Azure cloud services for the scenario of large-scale IoT devices. 

Some our comments are listed below:

1. Authors shall describe their unique contribution in a more clear way in Introduction (e.g., Sec. 1.2).

We know authors investigated the overall security integration for large-scale

IoT devices, but the technical contents seems a lot of implementation.

If they give more detailed explanation, this will increase their research value.

2. Figure quality (including font size, e.g., Fig. 2) can be improved. We only give an example. Authors shall check all figures and revise unqualified ones.

3. Author developed programs for MT3620 and CC2530 ZigBee devices. Can authors give a general

development principle consiering overall security integration for large-scale IoT devices?

Author Response

Response to Reviewer 3 Comments

We thank the reviewers for their constructive comments, which have helped improve the quality of the manuscript. The comments are well taken and the manuscript has been revised accordingly. In the following, we provide an itemized response to the comments and questions raised by the reviewers. For the reviewers’ convenience, their original comments are copied here and shown in italics, and our responses are highlighted in red.

Point 1:

Authors shall describe their unique contribution in a more clear way in Introduction (e.g., Sec. 1.2).

We know authors investigated the overall security integration for large-scale IoT devices, but the technical contents seem a lot of implementation. If they give more detailed explanation, this will increase their research value.

Response 1: We greatly appreciate the honourable reviewer for the constructive suggestion. In the revision, we modify the title of this article from “The Integration of Azure Sphere and Azure Cloud Services for Security-Oriented Internet of Things” to “The Integration of Azure Sphere and Azure Cloud Services for Internet of Things”. We make clear that this article is mainly focused on the integration solution for IoT application. And then we rewrite the instruction section to point out the challenges existing in the current IoT integration solutions. Afterwards, the contributions are listed in the form of bullet point so that it is clear to read. In the rest of this article, we justify with knowledge and practice that the high security, low cost and easy device management would be ensured via proposed integration.

Point 2:

Figure quality (including font size, e.g., Fig. 2) can be improved. We only give an example. Authors shall check all figures and revise unqualified ones.

Response 2: We greatly appreciate the honourable reviewer for the detailed suggestion. In the revision, we revise the unqualified figures such as Figure 1, Figure 3, Figure 4.  Furthermore, Figure 9 and Figure 11 are added to show the ZigBee data captured by Packet Sniffer and Azure IoT Hub data captured by Device Explorer Twin.

Point 3:

Author developed programs for MT3620 and CC2530 ZigBee devices. Can authors give a general development principle considering overall security integration for large-scale IoT devices?

Response 3: We greatly appreciate the honourable reviewer for the constructive suggestion. In the revision, we rewrite the system security considerations in Subsection 3.5. And the security considerations are concluded in Table 2 on page 8. In this article, the security policies and rules are divided into four categories, i.e., device security, connection security, cloud security and application security. Along with device security, connection security and cloud security, application security is added in the revision according to the honourable reviewer’s suggestion. Specifically, the designed solution achieves 128-bit AES algorithm for data encryption between the CC2530 ZigBee nodes. Furthermore, application capabilities, device capabilities and signing deployment are introduced to ensure the application security running on MT3620.

Round 2

Reviewer 1 Report

First of all, congratulations on the effort made to answer all my comments. I think the organization of the paper is quite better as well as the clearance of the contents. There are some aspects, especially regarding the implementation and results that could be better but I consider that the paper has already enough quality to be accepted.
However, there are a lot of phrases which English could be much better. On my comments, i left some suggestions as well as some sentences to enhance. The authors don't need to highlight the changes. The important now is to have a clear paper.
There is a lot of places where the authors wrote integration when I think they should write integrated. I did some corrections until chapter 2 but I gave up after that. Authours should correct the rest. 

Revisiting the paper: 

0.      Abstract: Much clearer even though it is not perfectly clear the novelty of the paper.

Line 9: integration -> integrated

Line 13: which -> that

Line 20 e 21: on the software and services other than end devices -> on the software and services rather than on the end devices

Line 21: integration -> integrated

Line 22: way for security -> way to ensure security that …

Line 38: delete integration

1-      Introduction: Much clear and objective.

Line 43: vehicle -> vehicles ; smart city -> smart cities;

Line 50: first -> firstly ; Line: 56: secondly; Line 59: thirdly

Line 57: very few of the existing integration solutions ->few existing solutions considered the security aspects all the way from the device to the cloud.

Line 59: overall security from system level -> overall system security.

Line 62: integration -> integrated; from the device layer to the cloud layer.

Line 63: integration -> integrated solution that includes Azures Sphere devices and Azure cloud services

Line 66: maintaining the low-cost requirement.

Phrase of line 67 needs to be re-written. Maybe: Moreover, a proof of concept application (a remote monitoring and feedback control application) is detailed, namely its design and implementation to validate the proposed integrated solution.

Line 71: are as follows -> are the following

Line 72: integration -> integrated;

Line: 75: which are acted as direct -> which act either as direct a directed connected device or a gateway, are designed and implemented based ….

Line 82: structed -> structured

Line 83: IoT integration solutions in the current studies are …-> IoT integrated solutions is reviewed.

Line 84: the implementation details which are comprised of hardware -> the implementation details that comprise hardware ….

Line 86: from aspects ..-> concerning the device layer …

2-     State of the art: It’s ok. On future works, the authors shall be more careful doing the state of the art. It is important to have more details on related work. Moreover, relating Azure Sphere, if I understood well, the authors are providing an evaluation of the refereed platform considering a particular use case. It is the first?

Line 94 – 96: The English could be better. “… Two main research topics involving IoT system’s security;

Line 95: delete study; Line 96: solution -> solutions

Line 98: model -> models

Line 97 – 116: I think you could maintain the firstly , secondly… lastly… It is better than first, second… lastly.

Line 118: A comparison of several surveyed papers is summarized in table 1.

Line 140. Focused on the implementation of an integrated solution that could ensure security on a IoT system, from the low-cost IoT device to the applications and services deployed in the cloud.

Line 145: was released

Line 149: to incorporate high levels of security into…

Line 152: were identified by Microsoft, namely: hardware-based root of trust, ….

Line 154: motivated by refereed goals, Azure Spere MCO, Azure, Sphere OS and Azure Spere Security Service were designed to …

Line 157: comprises the following three main components

Line 158- 167: you don’t need the firstly, secondly… you are already using bullets..

3-     System design: It’s ok. Now we can find a more general introduction of the system before entering on a particular case.

First phrase: needs to be re-written;

Line 203: The system is composed of three fundamental modules: i) the device connectivity…. ii)   …. ; and iii) ….

Line 205: delete The features of each module….

Lines 207-216: needs to be re-written. The english is not ok.

Line 246: which allows the registration and connection of large sets of …

Line 248: authors don’t state this azure function on the introductory phrase. Is it part of what ? business logic?

Line 255: it is not necessary bullets to state only two topics. Authors can put it all together.

Line 269: The MT3620 development kit, powered by rechargeable lithium-ion batteries, is designed 269 to perform as direct connect device or gateway device

Line 270: to act as a direct connected device or a gateway for legacy nodes

Line 271: sound sensor and a relay, are connected via…

Line 272: And direct HTTP ??

Line 281: THe authors don’t need to write always “in this article”. Consider removing it. “The services provided by Azure cloud, play a key role in data collection,” …

Lne 285: The Azure IoT Hub plays a central role. It acts as a bridge between the Azure Sphere devices and …

Line 293: again “in this solution”…

Line 297: filtering

Line 306: again “in this project”

Line 309: once the editing reports is finished,

Line 312: again “in this project”

Line 320: with a suitable ….For a lower latency, the closest location shall be selected.

Line 325: delete de second certificate

Line 332: that was validated

Line 339-341: In order to ensure security through all layers of the IoT system, the security policies and rules were divided into four categories as shown in Table 2.

Line 371: which is considered one of the strongest …. (should have a reference)

4-     System implementation: it's a bit poor but ok. 

Line 385: remove integration

Line 387: remove as follows

Line 413: as direct connected device

Line 419: remove In this article and replace by for example: On the system implementation, Zigbee end device is..

Line 421 temperature and humidity sensors

Line 424-426: it is a repetition of what was written before. Complete the first sentence and put a reference to the table.

Line 428: has no ADC and does not support I2C communications;

Line 433: …wireless network topology, that are presented in Table 4.

Line 436: IEEE 802.15.4, the basis of ZigBee protocol (add a reference here).

Line 441: The coordinator …

Line 443: Remove the And; Line 447: remove the And.

Line 470: for the device layer, including the MT3620 device and …

Line 473: Remove Necessary  

Line 484: via onboard interfaces

Line 495: remove And. Finally, at the end of the referred processes, the message is transmitted to Azure IoT Hub

Line 496: direct methods on the devices

Line 497: and can be called as soon as a ..

Line 500: the response is encapsulated immediately on the message payload and sent as a reply to the previous callback function.

Line 503: The MT3620 can also be configured as a gateway to allow legacy ioT systems to connect to the Internet and to the IoT Sphere IoT Hub.

Line 504: delete In this article.

Line 515: The flowchart of the application that is running in the …

Line 517: peripheral is configured and opened for inbound…

Line 520: as the transmission timer… sensor network is sent to the

Line 530: (OSAL) executes the main loop (don’t use future on this paragraph). There are several situations. Please correct them.

Line 549: again in this article….this paragraph don’t say nothing new.. The authors just write that the details were already detailed (I would suggest to delete it). The same for the first phrase of the paragraph that starts on line 553.

Line 554: The entities of the database are shown in table 6..

Line 563: Furthermore, the dataset becomes available in user’s …

5-     Experimental results: also a bit poor.  

Line 600-613 and 614 – 619. Badly written. Needs to be re-written. Figure 9 it is not explained clearly.

Line 633 – 634: I left to the authors the option to maintain or not the UART part. For me, it doesn't have any value.. at least considering the way it is done. Nevertheless, if the authors want to maintain it, please re-write the lines 633-634.

Line 662: it is streamed.

Line 673: remove as below

Lines 692- 696 – repeated. Consider removing it

6.      Discussion

Line 707: user scenarios mostly consist of sensors and actuators…

Line 718: much of those works are devoted .. is this phrase correct ?

7. Conclusion

Last phrase is not ok .

Please read all the paper carefully and try to enhance the English component. 

Reviewer 3 Report

Authors have improved the manuscript according to reviewers' comments.

Author Response

This manuscript is a resubmission of an earlier submission. The following is a list of the peer review reports and author responses from that submission.

Round 1

Reviewer 1 Report

Although the architecture and the experiments are quite interesting, the article mainly demonstrates how different Microsoft solutions can be implemented to create a secure architecture for large-scale scenarios. However, it is unclear what the actual contribution of their work is. The authors indicate three contributions, and all support my view of this project mainly being an integration of Microsoft solutions but nothing else. I would suggest the authors to rethink those contributions, is it that they wrote code to integrate them? Did they develop a new protocol? Is it a new strategy for integration?

Another issue with the article is that, although they mention security throughout, there is no clear indication on what the proposed architecture is actually securing. Is it the communication between MT3620 and the IoT Hub? Where exactly are the attack vectors that the authors are trying to solve? Is it just to secure communication? The article seems to focus more on performance.

Finally, most of the experiments, as indicated in my previous paragraph, focus on performance and/or on demonstrating that the integration works and real-time information can be collected. However, if the main focus of the articles was specified in terms of security then the experiments should also focus on that. The authors seem to trust in the security offered by the Microsoft solutions, which is is completely valid, but then the title should be changed as well as the introduction and other sections that discuss security. 

Author Response

We thank the reviewers for their constructive comments, which have helped improve the quality of the manuscript. The comments are well taken and the manuscript has been revised accordingly. In the following, we provide an itemized response to the comments and questions raised by the reviewers. For the reviewers’ convenience, their original comments are copied here and shown in italics, and our responses are highlighted in red.

Point 1: Although the architecture and the experiments are quite interesting, the article mainly demonstrates how different Microsoft solutions can be implemented to create a secure architecture for large-scale scenarios. However, it is unclear what the actual contribution of their work is. The authors indicate three contributions, and all support my view of this project mainly being an integration of Microsoft solutions but nothing else. I would suggest the authors to rethink those contributions, is it that they wrote code to integrate them? Did they develop a new protocol? Is it a new strategy for integration?

Response 1: We greatly appreciate the honourable reviewer for the detailed suggestion. In the revision, we rewrite the “introduction” section, add the summary of the current security considerations for IoT integration solutions in different layers (please refer to Table 1 and the second paragraph on page 2), point out the two challenging security issues in current IoT solutions (at the end of the second paragraph on page 2), present an overview of Azure Sphere and the related services that can be a promising solution to address the two challenging problems (in section 1.1 on page 3), reconsider and rewrite the contributions of this article (in section 1.2 on page 4). As an aside, we should emphasize that the main contribution is a security-oriented strategy for the integration of hardware, software and services that is based on Azure Sphere device and Azure cloud services. Then we design and construct the hardware prototypes, write applications for MT3620, CC2530 ZigBee and Windows devices, write and config services that running in the cloud, establish and verify the testbed of the proposed integration.

Point 2: Another issue with the article is that, although they mention security throughout, there is no clear indication on what the proposed architecture is actually securing. Is it the communication between MT3620 and the IoT Hub? Where exactly are the attack vectors that the authors are trying to solve? Is it just to secure communication? The article seems to focus more on performance.

Response 2: We consider this comment is of great significance to our study. In the revision, we have followed the honourable reviewer’s suggestion and managed to analyse the security policies and rules used in this work. We add “Section 2.2 System Security” on page 5 for the demonstration of security that starts in the device and extends to the cloud. The security policies and rules are listed in Table 2, which are categorized as device security, connection security and cloud security. And the security policies and rules are explained in detail from the first paragraph to the third paragraph right below Table 2 on page 5.

Point 3: Finally, most of the experiments, as indicated in my previous paragraph, focus on performance and/or on demonstrating that the integration works and real-time information can be collected. However, if the main focus of the articles was specified in terms of security then the experiments should also focus on that. The authors seem to trust in the security offered by the Microsoft solutions, which is completely valid, but then the title should be changed as well as the introduction and other sections that discuss security.

Response 3: We greatly appreciate the honourable reviewer for this constructive comment. The main contribution of this work is the strategy for the integration of hardware, software and services that is based on Azure Sphere device and Azure cloud services for IoT security applications. We find that this new integration can be expected to address the two challenging problems (in section 1.1 on page 3) that the existing solutions have not been solved yet. To this end, we modify the title from “A Security Solution Using Azure Sphere and Azure Cloud Services for Internet of Things” to “The Integration of Azure Sphere and Azure Cloud Services for Security-Oriented Internet of Things”, rewrite the “introduction” section, add the summary of the current security considerations for IoT integration solutions in the second paragraph on page 2, compare the security considerations with the existing IoT solutions in Table 1 on page 2, add a section “4.4 Discussion” at the end of section 4 on page 15 to further clarify the advantages that this work provides.

Reviewer 2 Report

This manuscript provides a security solution including Azure Sphere and other components at different IoT layers. however, there are a set of issues that need to be addressed before being ready for publication.

1.    Contribution is weak. I would recommend  that authors need to acknowledge authors contributions after integration Azure Sphere into their security solution and before.

2.      where is the research methodology employed for achieving this work.

3.    I would recommend to write about the problem authors try to solve. Need to give more details.

4.     No mention to related work and related solutions for the same problem and how their work can be compared against.

5.    I would recommend to use a section to provide an overview of Azure sphere, which is the main component of the article

6.     How your results can be compared to existing solutions.

7.     How the proposed solution can be evaluated against real security threats and system response to it.

8.     Experimental result section is not clear enough to provide the results and authors discussion about it.

9.     Some grammar Errors need to be reviewed by a native English speaker such as in line 123 and Line 131.

10.  Most provided URLs in references are not working 

Author Response

We thank the reviewers for their constructive comments, which have helped improve the quality of the manuscript. The comments are well taken and the manuscript has been revised accordingly. In the following, we provide an itemized response to the comments and questions raised by the reviewers. For the reviewers’ convenience, their original comments are copied here and shown in italics, and our responses are highlighted in red.

Point 1: Contribution is weak. I would recommend that authors need to acknowledge authors contributions after integration Azure Sphere into their security solution and before.

Response 1: We greatly appreciate the honourable reviewer for the detailed suggestion. In the revision, we rewrite the “introduction” section, add the summary of the current security considerations for IoT integration solutions in the second paragraph on page 2, compare the proposed Azure Sphere based solution with the existing IoT solutions in Table 1 on page 2, reconsider and rewrite the contributions of this article (in section 1.2 on page 4). As an aside, we should emphasize that the main contribution is a security-oriented strategy for the integration of hardware, software and services that is based on Azure Sphere device and Azure cloud services. Hence, we modify the title from “A Security Solution Using Azure Sphere and Azure Cloud Services for Internet of Things” to “The Integration of Azure Sphere and Azure Cloud Services for Security-Oriented Internet of Things”.

Point 2: Where is the research methodology employed for achieving this work.

Response 2: We greatly appreciate the honourable reviewer for this constructive comment. In the original manuscript, the research methodology is included in “Section 3 System Design and Implementation”. In the revision, we add “Section 2.2 System Security” on page 5. The security policies and rules are listed in Table 2, which are categorized as device security, connection security and cloud security. And the security policies and rules are explained in detail from the first paragraph to the third paragraph right below Table 2 on page 5.

Point 3: I would recommend to write about the problem authors try to solve. Need to give more details.

Response 3: We greatly appreciate the honourable reviewer for this helpful comment. Accordingly, in this revision, we have further provided the related references for two main fields of security research in the existing work (in the second paragraph and the third paragraph in section 1 from page 1 to page 2). Then we find that there still remain two challenging issues in real IoT solutions. The first one is that very few of the existing solutions have considered the security all the way from the device to the cloud. And the second one is that the need to incorporate the high-value security into every low-cost (much cheaper than 10 US dollars) network-connected IoT devices is currently underestimated because of the limited development costs and device capabilities (at the end of the second paragraph on page 2).

Point 4: No mention to related work and related solutions for the same problem and how their work can be compared against.

Response 4: We greatly appreciate the honourable reviewer for this helpful comment. To some extent, the existing studies have achieved the security and privacy goals in one specific layer or multiple layers. However, to the best of our knowledge, very few of the existing solutions have considered the security all the way from every low-cost device to the cloud simultaneously.

Point 5: I would recommend to use a section to provide an overview of Azure sphere, which is the main component of the article.

Response 5: We greatly appreciate the honourable reviewer for this constructive comment. We add an overview of Azure Sphere and the related services that can be a promising solution to address the two challenging problems (in section 1.1 on page 3). The hardware architecture and the related security services of Azure Sphere are presented respectively.

Point 6: How your results can be compared to existing solutions.

Response 6: We greatly appreciate the honourable reviewer for this constructive suggestion. Accordingly, the security considerations in this work are compared with the existing solutions in Table 1 on page 2.

Point 7: How the proposed solution can be evaluated against real security threats and system response to it.

Response 7: We greatly appreciate the honourable reviewer for this helpful comment. In this work, the integration of the available devices and services for large-scale scenarios with limited budgets, rather than the creation of security policies and rules is concentrated. Hence, we provide the integration in detail including hardware prototype, software design and cloud integration in section 3 and section 4. And the system response to the real security threats is not considered in this work.

Point 8: Experimental result section is not clear enough to provide the results and authors discussion about it.

Response 8: We greatly appreciate the honourable reviewer for this helpful suggestion. In this revision, we add the session “4.4 Discussions” at the end of the section 4 on page 15. We emphasize that the main contribution of the work is the integration of hardware, software and services that is based on Azure Sphere device and Azure cloud services to achieve security all the way from every low-cost device to the cloud simultaneously. We also further clarify the advantages that this work provides.

Point 9: Some grammar Errors need to be reviewed by a native English speaker such as in line 123 and Line 131.

Response 9: We greatly appreciate the honourable reviewer for this helpful comment. In this revision, some grammar errors in the manuscript have been revised accordingly. For example, the mistakes in line 94, line 99, line 123, line 131 and line 363 in the original manuscript are corrected.

Point 10: Most provided URLs in references are not working.

Response 10: We greatly appreciate the honourable reviewer for this helpful comment. In this revision, we check all the URLs in the original manuscript and the broken URLs have been revised. For example, URLs in reference 26, reference 27, reference 28, reference 29, reference 30, reference 32 and reference 34 in the original manuscript are corrected.

Round 2

Reviewer 2 Report

The manuscript in its new form is much better in term of readability and contribution discussion.

The authors to succeeded address my concerns. However, I recommend minor revision. I would recommend to have a separate section (Discussion) to discuss the contribution of the manuscript.  Also, I would recommend to Table 1 into the discussion section, since it will be more logical to compare your work with other people work after presenting your results.